# Data–driven modelling makes quantitative predictions regarding bacteria surface motility

**Daniel L. Barton**[1], **Yow-Ren Chang**[2], **William Ducker**[3], **Jure Dobnikar**[1,4,5]*

**1** CAS Key Laboratory of Soft Matter Physics, Institute of Physics, Chinese Academy of Sciences, Beijing, China, **2** National Institute of Standards and Technology (NIST), 100 Bureau Dr, Gaithersburg, Maryland, United States of America, **3** Department of Chemical Engineering and Center for Soft Matter and Biological Physics, Virginia Tech, Blacksburg, Virgina, United States of America, **4** Wenzhou Institute of the University of Chinese Academy of Sciences, Wenzhou, China, **5** School of Physical Sciences, University of Chinese Academy of Sciences, Beijing, China

* jd489@cam.ac.uk

**Data Availability Statement:** All code written in support of this publication is publicly available at https://github.com/dluke/tmos. The experimental tracking data that were used in this work are

## Abstract

In this work, we quantitatively compare computer simulations and existing cell tracking data of *P. aeruginosa* surface motility in order to analyse the underlying motility mechanism. We present a three dimensional twitching motility model, that simulates the extension, retraction and surface association of individual Type IV Pili (TFP), and is informed by recent experimental observations of TFP. Sensitivity analysis is implemented to minimise the number of model parameters, and quantitative estimates for the remaining parameters are inferred from tracking data by approximate Bayesian computation. We argue that the motility mechanism is highly sensitive to experimental conditions. We predict a TFP retraction speed for the tracking data we study that is in a good agreement with experimental results obtained under very similar conditions. Furthermore, we examine whether estimates for biologically important parameters, whose direct experimental determination is challenging, can be inferred directly from tracking data. One example is the width of the distribution of TFP on the bacteria body. We predict that the TFP are broadly distributed over the bacteria pole in both walking and crawling motility types. Moreover, we identified specific configurations of TFP that lead to transitions between walking and crawling states.

## Author summary

Twitching is a type of bacterial surface motion facilitated by molecular motor-driven micron-scale filaments known as type IV pili (TFP). The resulting motion, which is characterised by erratic behaviour at short timescales, is important for bacterial surface invasion, biofilm formation etc. The link between microscopic mechanisms on the level of single filaments and macroscopic properties of twitching trajectories is generally difficult to make, since direct time-resolved imaging is of TFP in experiments is challenging. We propose a data-driven model for surface motility of the bacteria P. Aeruginosa that bridges these scales: we resolve single-TFP dynamics and connect it to the macroscopic properties of twitching trajectories. We quantitatively compare our simulations to experimental

available at https://github.com/dluke/plos_tmos_data.

**Funding:** The work was supported by the Chinese National Science Foundation through the grants 11874398, 12034019, by the Strategic Priority Research Program of the Chinese Academy of Sciences through the grant XDB33000000, and by an international collaboration grant from the K. C. Wong Educational Foundation. The funders had no role in study design, data collection and analysis, decision to publish, or preparation of the manuscript.

**Competing interests:** The authors have declared that no competing interests exist.

tracking data, predict the previously unresolved distribution of TFP on bacterial membrane, and discuss how it affects the transitions between the "walking" and "crawling" modes of motion. We also extract the speed of TFP retraction corresponding to the specific experimental conditions, which is lower than reported in several other experiments, but in perfect agreement with measurements performed at conditions of our tracking data. Our work thus elucidates microscopic mechanism of twitching, and it represents a systematic data-driven approach to quantitative modelling of complex biological phenomena.

# 1 Introduction

Many pathogenic bacteria form surface associated colonies that are resistant to immune system agents and antibiotics. This colonisation is aided by their capability to actively propel themselves on surfaces. Surface motility of *P. aeruginosa* and other bacteria is facilitated by molecular motors which drive extension and retraction of micron-scale filaments known as type IV pili (TFP). The resulting motility, characterised by its erratic behaviour on short timescales, is known as *twitching*. The link between surface motility and TFP was first established for the pathogenic *P. aeruginosa* bacteria by observing the spreading behaviour of growing colonies [1]. Subsequently, motility has emerged as a common factor in bacterial pathogenicity [2], prompting interest in the motility apparatus as a target for treating infection. Structural studies show that TFP are mainly composed of copies of a protein subunit PilA which polymerize into a stiff helical filament with a diameter of approximately 5 nm [3].

Our understanding of the extension/retraction behaviour of TFP and their mechanical properties during twitching is greatly enhanced by observation using fluorescence microscopy [4, 5] and interferometric scattering microscopy [6], which enable measurements of the extension and retraction rates and the persistence length of TFP. A single extension–retraction cycle is typically seen to take a few seconds, however a clear understanding of TFP behaviour can only be obtained by observation of many TFP cycles over a period of many minutes and there is yet more to learn on longer timescales regarding how TFP dynamics influence surface detachment [7], transitions in orientation [8], trajectories on patterned surfaces [9, 10], and in the presence of obstacles [11–13], or other biological stresses [14]. The mechanical and retractile properties of individual TFP on short timescales has also been measured indirectly using atomic force microscopy [15], optical tweezers [16–18] and force sensitive micro-pillar assays [19].

Live cell imaging of twitching motion with visible TFP over longer timescales is challenging. Hence, motility of *P. aeruginosa* has been extensively studied by single-cell tracking methods where the bacteria position can be detected but not the state of its TFP. Typical tracking methods involve fluorescence microscopy followed by image segmentation. These methods scale well to the simultaneous tracking of hundreds of bacteria over periods of several hours and are well suited to the study of biologically relevant processes such as surface colonisation and early stage biofilm formation [20]. In this way, *P. aeruginosa* have been seen to display a variety of TFP and flagella driven motility types [7]. Most notably, a directionally-persistent 'crawling' motion with both poles in contact with the surface and a low-persistence 'walking' motion with one pole in contact [21]. Multi-generational tracking has revealed that *P. aeruginosa* populations retain memory of interacting with a surface much longer than their division timescale. This capability appears to facilitate various cooperative strategies for surface exploration and colonisation [22, 23].

Despite the growing body of knowledge regarding the microscopic structure and physical properties of TFP, their dynamics still cannot be resolved in long-time twitching trajectories, and the arrangement of TFP and their coordination on macroscopic length- and timescales are not well understood. Considering the wealth of available single-cell tracking data it is tempting to use this information combined with *in-silico* modeling in order to infer aspects of TFP behaviour that are inaccessible to modern imaging techniques. Indeed, *P. aeruginosa* in the crawling state has been seen to exhibit intermittent, short and fast displacements, coupled with a rotation of the body, which are interpreted to be caused by the release of a single TFP while the bacteria is in a jammed configuration of multiple surface-bound TFP [21]. This interpretation is supported by a two dimensional twitching model [24] that is calibrated by *P. aeruginosa* tracking data. A simulation approach has also been used to study the qualitative properties of the twitching motility of bacteria *N. gonorrhoeae*, both individually and in dense colonies by explicit modelling of pili-surface and pili–pili interactions [25].

In this work, we construct a three-dimensional physical model of TFP–driven twitching dynamics. We quantitatively compare our simulated trajectories to experimental trajectories using approximate Bayesian computation [26]. We are able to obtain estimates for characteristics of *P. aeruginosa* TFP that are difficult or impossible to measure directly in experiments, such as the TFP distribution on the cell surface. We also demonstrate mechanisms by which *P. aeruginosa* may use TFP activity and polarization to drive transitions between walking and crawling states.

## 2 Modelling

We model the bacterium as a rigid sphereocylinder with radius $r = 0.5$ μm and length $l = 3.0$ μm, which are typical dimensions for *P. aeruginosa* [8]. One pole of the spherocylindrical body supports the growth of TFP [4], which are anchored to the bacteria membrane at one end while the other end is free to interact with the environment. During the simulation, at each time step TFP may either extend, retract, change their extension/retraction state, or change their surface binding state. Free TFP are modelled as worm-like chains and do not influence the dynamics of the body, but become surface-bound at the moment the chain intersects the surface. Surface-bound TFP drive motion of the cell by retracting. On the other hand, we assume that surface-bound TFP cannot exert forces significant enough to push the cell by extending. For simplicity, we assume that TFP may only bind the surface at their tips, although more complex surface interaction patterns have been proposed [27].

Due to the asymmetry of the bacteria, TFP pulling on one pole exert a torque on the body, which favours an upright orientation where only the leading pole is touching the surface, *i.e.*, a "walking" configuration. A result of this torque is that a significant attractive body–surface interaction is necessary to maintain the crawling state. The majority of the bacteria in the experimental tracking data that we analyse [14] are crawling for the entire duration of the measurement, which suggests these bacteria stick firmly to the surface. Skerker & Berg also argue that the surface interaction between *P. aeruginosa* and the glass coverslip must be quite strong in their experiment [4], because many of the bacteria they observed did not move, despite showing TFP activity. Since bacteria in this data set rarely transition between crawling and walking states (see S9 Appendix), we deliberately study crawling and walking behaviour separately both in simulations and experiment until section 5. In order to simulate both walking and crawling modes of motion, the cells are decorated with two interaction sites (one at each pole, see Fig 1B). The short range interaction of the two sites with the surface is either purely repulsive, or an attractive well (see S1 Appendix for details). A purely repulsive surface interaction is used to simulate walking motility while crawling motility is simulated by initialising

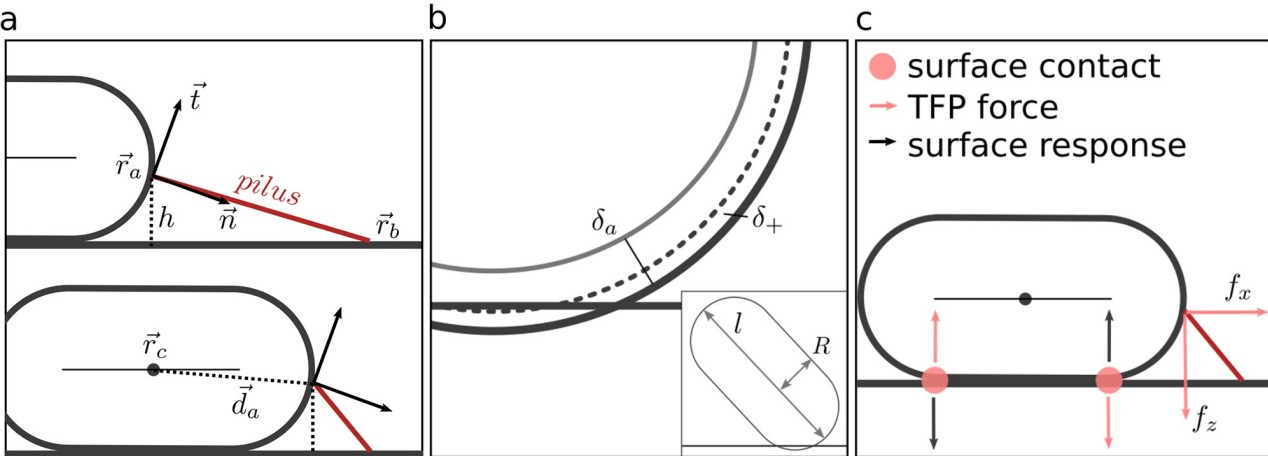

**Fig 1. Model schematics.** (a) Side view of a spherocylindrical body on a flat surface with a single TFP attached to the surface in the retracting mode. TFP retraction drives the forward motion to the state in the bottom panel. Vectors $r_a$ and $r_b$ define the positions where TFP are anchored to the cell and attached to the surface, respectively, while $n$ and $t$ are the surface normal and tangent at $r_a$. (b) Surface interactions are modelled by one contact point at each pole. The $z$ component of the TFP retraction force pulls the leading pole down and forces the trailing pole up. Black arrows indicate the direction of the restoring forces due to a potential well at the surface. (c) Rendered snapshot of a simulated walking bacteria with several surface bound TFP (red) and several free TFP (blue).

both poles in contact with the surface and using an attractive potential to maintain the horizontal orientation of the cell. This approach avoids the need for more detailed modelling of the bacteria-surface interactions which are poorly understood and may vary between experiments and between individual bacteria.

Direct measurements of the typical friction and viscous forces on twitching bacteria are very scarce and we do not attempt to model these dissipative forces explicitly, instead, they enter the model indirectly through the TFP retraction speed. We assume that the molecular motors are sufficiently strong to produce pulling forces incurring tension in the TFP that can balance the sum of the dissipative forces. The fact that the retraction speed of *P. aeruginosa* TFP is reduced by the tension incurred from dragging the cell is supported by both direct observation of twitching [4, 28, 29] and single pilus experiments [30]. To incorporate these findings, we treat the TFP retraction speed as a free parameter that is optimized in our data-driven modelling to best adapt to the conditions at which the experimental tracking data was obtained. Where multiple TFP pull against each other, they still retract at a constant rate until the pilus tension reaches the stall force, at which point the retraction stops. For more realistic models of bacteria–surface interactions, new experimental tools to measure the tension force in surface-bound TFP during twitching of live bacteria, as well as the surface friction and adhesion forces would be invaluable.

## Kinetic Monte Carlo simulations

The time evolution of the system is simulated using a Kinetic Monte Carlo (KMC) procedure [31], as a series of consecutive single-TFP events selected according to a corresponding set of rates that are defined as model parameters. Following the single-TFP event, an energy relaxation step is performed that adjusts the position of the cell body such as to balances the TFP tension and surface interaction forces. This rigid-body optimisation is described in detail in S2 Appendix.

TFP are in the extension or retraction state if they are bound to a corresponding motor [5]. In our simulations, extension and retraction of TFP are discretised using a small step size of $\delta_{\text{step}}$ = 4 nm. Each TFP exclusively binds either an extension or to a retraction motor with the rates $k_{\text{ext,on}}$ and $k_{\text{ret,on}}$, respectively. The same motors unbind with rates $k_{\text{ext,off}}$ and $k_{\text{ret,off}}$. New TFP are created with a spawn rate $k_{\text{spawn}}$ and always start with the extension motor attached. TFP are dissolved immediately upon their retraction to zero length.

Table 1 summarizes the model parameters (*i.e.*, event rates) of our model. Some rates have been measured in experiments and when relevant estimates are possible, we list the values in the table. Measurements of the persistence length of TFP vary widely [27, 32], we list the value $L_p$ = 5.0 μm [4] in the table and discuss this again in section 4. The membrane-spanning TFP machine that constructs, destructs and stabilises TFP has been resolved by cryo-election tomography in the bacteria *Myxococcus xanthus* [33], and the Young's modulus of the machine as a whole is measured to be $E = 2 \times 10^3$ pN/μm [15]. The stall force of the retraction motor is reported to be approximately $f_{\text{stall}}$ = 100 pN [17, 34]. The $k_{\text{resample}}$ parameter represents an approximate coarse-graining of the thermal dynamics of unbound TFP. We consider $k_{\text{resample}}$ to be roughly proportional to the auto-correlation decay time of the spatial configuration of the TFP polymer. The rate 1.0 s$^{-1}$ is an order of magnitude estimate, given that these fluctuations have been observed by video microscopy [4].

The TFP extension and retraction speeds have been reported several times to be between 0.5 μms$^{-1}$ and 1.0 μms$^{-1}$ [4–6], although Skerker & Berg also measured the actual motion of the cell body to be $\sim$ 0.3 μms$^{-1}$ which suggests that the act of displacing the cell slows TFP retraction. TFP retraction speed while dragging the cell has recently been measured by Zhang et al. [28] at 0.09 μms$^{-1}$. We argue that these large experimental discrepancies are caused by different experimental procedures adopted in the various experiments. For example, a temperature change of 8˚ was shown to cause a 2× increase in retraction velocity [30]. In general, the effects of external conditions on twitching motility are not well explored. In addition, direct imaging of TFP during twitching is rarely available for the same conditions as used in large-scale tracking experiments. The tracking data used to calibrate our model (described in section 3), however, are obtained following nearly identical procedures to those used by Zhang et al. [28], meaning that we expect their recent measurements of the extension and retraction speeds to correspond well with the tracking data we study.

**Table 1. Model parameters with experimental estimates.** The subscript and superscript for the motor binding rates are the 95% confidence interval [5].

| Parameter | Estimate | Units | Description | Reference |
|---|---|---|---|---|
| $k_{\text{dwell}}^{-1}$ | 1.0 | s | TFP surface binding duration | Tala et al. [6] |
| $\kappa$ | - | - | Width of TFP Distribution | - |
| $k_{\text{spawn}}$ | - | s$^{-1}$ | Rate of TFP production | Koch et al. [5] |
| $k_{\text{resample}}$ | 1.0 | s$^{-1}$ | TFP resampling rate | - |
| $k_{\text{ext,off}}^{-1}$ | $1.6^{+0.5}_{-0.2}$ | s | Unbinding rate of the extension motor | Koch et al. [5] |
| $k_{\text{ret,off}}^{-1}$ | $9.1^{+9.7}_{-3.8}$ | s | Unbinding rate of retraction motor | Koch et al. [5] |
| $k_{\text{ext,on}}^{-1}$ | $2.4^{+1.8}_{-0.3}$ | s | Binding rate of extension motor | Koch et al. [5] |
| $k_{\text{ret,on}}^{-1}$ | $0.40^{+0.30}_{-0.05}$ | s | Binding rate of retraction motor | Koch et al. [5] |
| $v_{\text{ret}}$ | 0.09 | μms$^{-1}$ | TFP retraction speed | Zhang et al. [28] |
| $v_{\text{ext}}$ | 0.28 | μms$^{-1}$ | TFP extension speed | Zhang et al. [28] |
| $L_p$ | 5.0 | μm | TFP persistence length | Skerker & Berg [4] |
| $E$ | $2 \times 10^3$ | pN/μm | Elastic modulus of the TFP machine | Beaussart et al. [15] |
| $f_{\text{stall}}$ | 100 | pN | Stall force of the retraction motor | Maier et al. [17] |

Where we list estimates for the remaining parameters, it is unclear whether they are intrinsic properties of *P. aeruginosa* or whether they vary depending on experimental conditions and between individual bacteria. The surface dwell time $\tau_{\text{dwell}} = k_{\text{dwell}}^{-1}$ has been seen to vary between $\sim 1.0$ s for the flagella deficient mutant and $\sim 2.3$ s for wild type [6]. It is unclear whether tension triggers the unbinding of TFP in *P. aeruginosa* as it does for *N. gonorrhoeae* [35] or not [6]. The spawn rate of TFP, $k_{\text{spawn}}$, varies from 0.1 s$^{-1}$ to 0.5 s$^{-1}$ within a population of *P. aeruginosa* in a single experiment [5]. An appropriate estimate for the $\kappa$ parameter, describing the width of TFP distribution around the leading pole has not yet been reported.

## Modeling TFP and force generation

Unbound TFP are modelled as discrete worm-like chains with persistence length $L_p = 5$ μm [4]. Instead of simulating the dynamics of flexible TFP, we approximately model their flexibility by regularly updating the configuration of pili using a Boltzmann generator (see S3 Appendix) with a re-sampling rate $k_{\text{resample}}$. If TFP intersect the surface, either by extension, movement of the body or by resampling, they become surface-bound and remain so until a detachment transition is chosen with a rate $k_{\text{detach}}$. A simplifying assumption is that TFP always bind immediately on their first contact with the surface. Bound pili are observed to be almost exclusively taut [4, 6] which leads us to model bound pili as elastic filaments, rather than worm-like chains.

Bound TFP are anchored to the body surface at position $\boldsymbol{r}_a$, defined relative to body center $\boldsymbol{r}_c$, by $\boldsymbol{r}_a = \boldsymbol{r}_c + \boldsymbol{d}_a$. Their ends are bound to the surface at position $\boldsymbol{r}_b$. The ideal length of the pilus is $l_{\text{eq}}$, while its actual length is $r_{ab} = |\boldsymbol{r}_b - \boldsymbol{r}_a|$, which incurs an elastic penalty

$$u^{\text{TFP}}(r_{ab}) = \begin{cases} \frac{E}{2} \frac{(r_{ab} - l_{\text{eq}})^2}{l_{\text{eq}}}, & r_{ab} < l_{\text{eq}} \\ 0, & r_{ab} \geq l_{\text{eq}}, \end{cases}$$

where $E$ is the elastic modulus. Bound TFP predominantly retract [6]. We assume that TFP retraction is the dominant driver of twitching motion and that TFP are not capable of pushing the cell by extension. Accordingly, no elastic energy is associated with TFP when $r_{ab} \geq l_{\text{eq}}$. If the position of the cell is stalled, *e.g.*, by the surface attachment of multiple TFP, then tension may build up in the filament until it is greater than the stall force, at which point TFP retraction is no longer possible.

A recent report by Tala et al. [6] concludes that the TFP machine has a surface sensing mechanism that triggers retraction rapidly after surface contact. Another recent work by Koch et al. [5] makes the claim that such a mechanism is not necessary to explain TFP behaviour. These reports lead us to investigate models with varying delay times between TFP surface binding and retraction (see S6 Appendix). We found that simulations with varying delay times could still produce trajectories that matched well to the experimental data. In fact the delay time had surprisingly little effect on the trajectory statistics overall. In these simulations, a larger delay time typically results in a larger average number of TFP, with the additional surface-bound TFP being in passive or extension states. The conclusion of S6 Appendix is that our analysis is not sensitive enough to provide evidence for or against the surface sensing mechanism. In this work we chose to proceed using a model with a short delay time, consistent with surface-sensing. Specifically, Tala et al. [6] measure the delay time between TFP binding to the surface and becoming taut to be $\tau_{\text{delay}} = 135$ ms. Since it appears that $\tau_{\text{delay}}$ is small compared to a single TFP extension/retraction/surface binding phase, we use the approximation $\tau_{\text{delay}} \to 0$ which implies that $1/k_{\text{ext,off}} + 1/k_{\text{ret,on}} \to 0$ for bound TFP. For unbound TFP, $k_{\text{ext,off}}, k_{\text{ret,on}}$ are model parameters for which experimental estimates are available. Due to the

relatively small effect of increased delay time on simulated trajectories (S6 Appendix), we expect many of the results in this work to be similar for alternative models that do not rely on surface-sensing.

Detachment transitions are accompanied by shrinking of the pilus by one unit of 4 nm. For taut pili, shrinking by 4 nm is enough to ensure that after detachment the pilus does not intersects the surface anymore. If the pilus is not taut at the moment we attempt to detach it, then it may intersect the surface after detachment and re-attach at the end of the simulation step. In this case, we repeat the shortening until it no longer intersects.

### TFP distribution

Experimental observations suggest that *P. aeruginosa* typically grows TFP on one pole during twitching [4, 6], however, a quantitative analysis of the TFP distribution on the cell surface has not been reported. In our model, we suppose that the TFP are distributed around the cell pole according to the von-Mises Fisher distribution, which resembles a Gaussian distribution on a spherical surface:

$$f_p(\boldsymbol{x}; \kappa) = \frac{\kappa}{4\pi \sinh \kappa} \exp\left(\kappa \hat{\boldsymbol{b}} \cdot \boldsymbol{x}\right), \tag{1}$$

where $\boldsymbol{x}$ is a point on the unit sphere, $\boldsymbol{b}$ is the long axis of the spherocylinder, and $\kappa$ is a parameter controlling the width of TFP distribution: smaller $\kappa$ means a broader distribution. A spherocylinder is composed of a cylindrical part with a spherical cap on each end. To accommodate TFP growth on the cylindrical portion of the body we use a linear transformation $g(\theta)$: $[\pi/4, \pi/2] \rightarrow [\pi/4, \pi/4 + l/2]$ to map points from the lower half of the sphere with radius $R = 0.5$, onto half of the cylindrical portion of the spherocylindrical body (Fig 2).

### Assumptions

For convenience, the key model assumptions are compiled and presented in Table 2, alongside some of the alternatives that were considered. The assumptions made in this work rely as much as possible on the state-of-the-art experimental information. Relaxing any assumption would lead to new classes of models that are variations of the one considered here. However, this only becomes useful if the experimental data and statistical analysis (sections 3, 4) are sufficient to distinguish between such models with a high confidence, and to achieve this, new experiments and improved imaging techniques are necessary. Several model choices, such as those related to surface sensing and the TFP distribution are particularly straightforward to adjust according to potential new experimental data. Our simulation codes are provided, with the view that the method we are presenting may be a useful reference for future studies.

## 3 Tracking data

To calibrate our model we obtained high–throughput tracking data of the flagella deficient ΔfliC mutant of the PA01 strain of *P. aeruginosa*. The flagella-less mutant is used to ensure that the observed motility is driven solely by TFP. The PA01 strain has been used in many experiments [5, 6, 14, 28, 30] although not by Skerker & Berg [4].

For completeness the experimental procedures used to obtain this tracking data are discussed here, for more details see the procedure described in [14]. The bacteria were initially grown on LB agar plates at 37° for 24 hours, monoclonal colonies were isolated and then incubated again in a culturing tube and then collected at the exponential phase. The resultant culture (20 $\mu$l) was diluted 50 times in 1 ml of FAB medium and injected into a flow cell. The FAB medium is minimal medium containing the essential components for cell growth, into which,

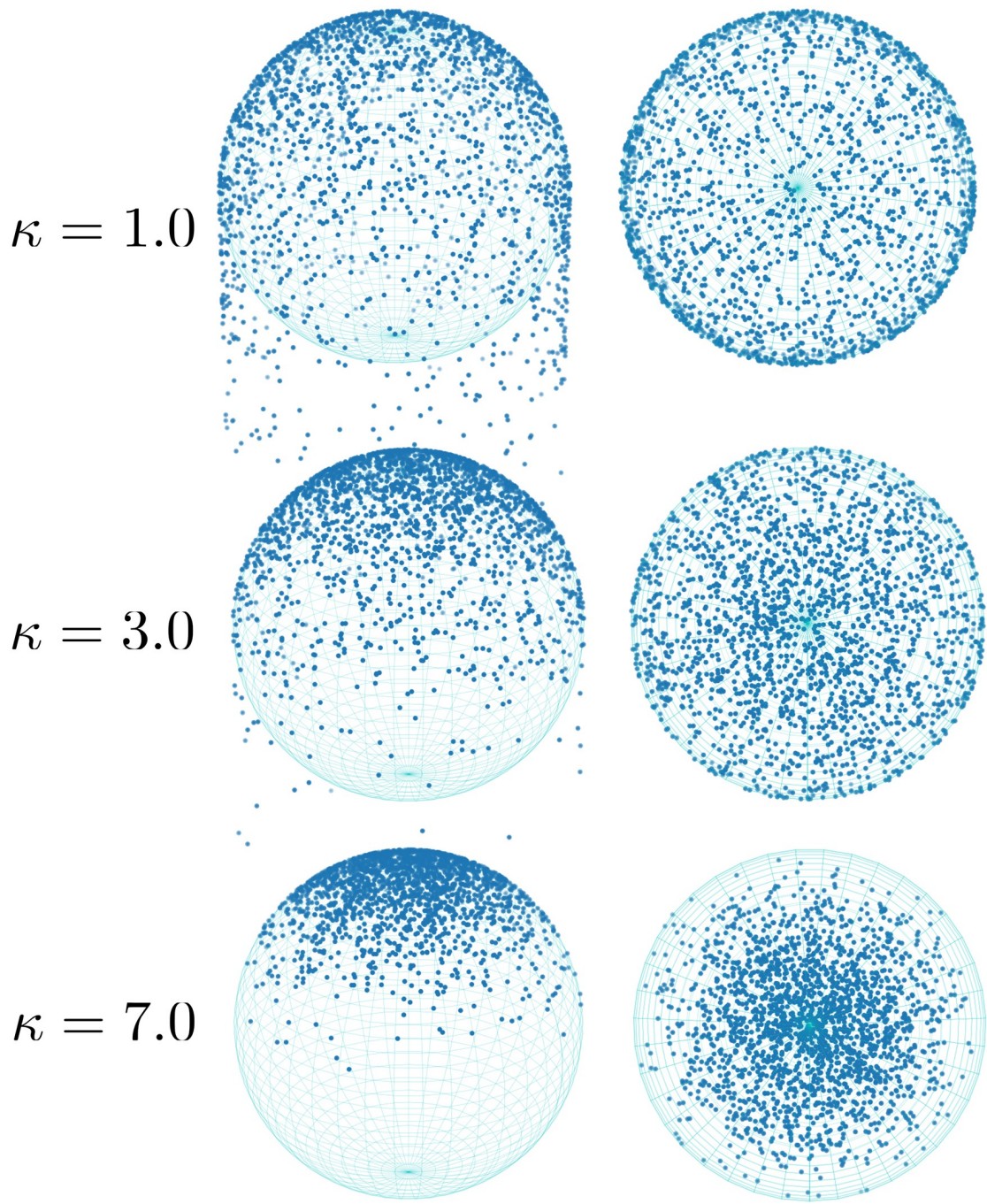

**Fig 2. Pili distribution.** Side view (left) and top view (right) of 2000 samples drawn from the TFP distribution for $\kappa$ = 1.0, 3.0, 7.0.

additionally, glutamate was added as a carbon source [36]. Bacteria were allowed to attach to the coverslip for 15 minutes, after which unattached cells were washed out. The remaining cells were maintained at a constant temperature of 30° in FAB media at a flow rate of 3.0 ml/h. At this time they were imaged for 2000 seconds at 10 fps using a microscope equipped with a 100× oil objective and a sCMOS camera (Andor Neo) with a pixel size of 6.5μm. The

**Table 2. Model assumptions and discussion.**

| Assumptions in our work / *Supporting evidence and discussion of alternatives.* |
|---|
| **TFP surface sensing** [6] / *We observed that the model is not sensitive to surface sensing* [5] (*see* S6 Appendix), *and consequently cannot be used to judge whether such mechanism exists.* |
| **TFP adhesion at tip only** / *More complex TFP surface interactions could be considered* [27] *but for simplicity we do not attempt to model them.* |
| **TFP Extension/retraction timescales are intrinsic to P. aeruginosa** / *We use experimental estimates of Koch et al.* [5] *to inform the model, even though experimental conditions vary between these experiments and the tracking data under investigation. Separately, we use sensitivity analysis to check which timescales strongly influence simulation results.* |
| **Tension-independent TFP unbinding timescale** / *The idea that high pilus tension triggers unbinding from surfaces was used to explain twitching of N. gonorrhoeae* [35], *however, there is some conflicting evidence as to whether this is a general feature of twitching motility* [6]. *For simplicity, the model TFP unbinding timescale is treated here as a tension-independent parameter.* |
| **Dissipative forces are not modeled** / *Force balance dictates that TFP pulling forces are balanced by dissipative forces, i.e., friction and viscosity. In the first approximation, these forces affect the motor activity and thus the TFP retraction speed, which is our model parameter.* |
| **TFP retraction speed approximated by a constant parameter** / *TFP retraction speed in P. aeruginosa was shown to be tension dependent* [30]. *Nonetheless for simplicity we use a constant retraction speed with reference to the retraction speeds reported in experiments* [4–6, 28] |
| **The von-Mises Fisher distribution can reasonably approximate the TFP arrangement** / *This is a generalized Gaussian distribution, which, without more specific information, is a natural choice to study the effect of the width of TFP distribution around the pole on the cell dynamics.* |
| **Out-of-plane rotations do not have a major effect in the crawling state** / *Due to the strong adhesion assumed in the crawling mode, this is a reasonable assumption. See* S1 Appendix *for the implementation of the bacteria–surface interaction.* |
| **Brownian forces are insignificant for crawling cells** / *Due to the strong surface adhesion in the crawling state, this is a natural assumption.* |
| **Brownian motion is insignificant for walking cells** / *We assume that most of the time during walking, the cells are attached to the surface; either by TFP or otherwise.* |

experiment was repeated 5 times and the trajectory data aggregated. As with Zhang et al. [28], the temperature was 30˚, which is similar to Skerker & Berg [4] (29˚) but is notably different from [5, 6] and [30] (37˚). *P. aeruginosa* colonies are known to deposit extracellular polymers (EPS) on the surface, however, we do not believe that a build up of EPS in this time frame has a significant affect on the trajectory data, for evidence see S8 Appendix.

Bacteria positions and orientations were extracted from image data by ellipse fitting, the trajectories of the leading and trailing poles were then extracted by particle tracking. A wavelet de-noising algorithm was used to remove high frequency noise that is associated with measurement errors by smoothing measurements from several sequential time steps into a coherent trajectory. We do not directly compare smoothed experimental data with simulated trajectories. Instead, we pre-process all data following the linearisation procedure proposed by Jin et al. [21], which is to coarse–grain trajectories according to a fixed distance $\delta_{\text{step}} = 0.12$ μm, chosen because it is significantly larger than the scale of the noise. The linearisation is performed independently for the leading $\{r_i^L\}$ and trailing $\{r_i^T\}$ poles by selecting indices $\{s_i\}$ from the 0.1 s resolution data so that the consecutive displacements in the trajectory $\{r_{s_i}\}$ satisfy $|r_{s_i} - r_{s_{i-1}}| \geq \delta_{\text{step}}$. The trajectory segment from $r_{s_{i-1}}$ to $r_{s_i}$ is then considered a straight line. All further analysis and summary statistics are computed from the linearised trajectory of the leading pole of the bacterium unless otherwise stated.

The population of bacteria in this data set have widely varying behaviour consistent with previous observations [7, 14]. We sort them by the mean velocity of their leading pole and

classify them by their orientation relative to the surface, either crawling (body axis parallel to the surface) or walking (body axis out-of-plane). While it is straightforward to obtain the orientation of the bacterium in the crawling state, accurately resolving the out-of-plane motion of walking bacteria is difficult. To identify walking and crawling trajectories we define the aspect ratio $b_t$ at time $t$ as the length divided by width of the body projected onto the plane. We then coarse grain $b_t$ using a sliding window of 200 frames (20 s) and take the minimum value for each trajectory, call it $b_{\min}$. The $b_{\min}$ distribution is bimodal with walking trajectories having values close to 1 (Fig 3A). We classify trajectories with $b_{\min} > 1.6$ as crawling (2742) while the remaining (371) trajectories are classified as walking.

Within these classifications, the trajectories still vary to a large degree. For example, after sorting the crawling trajectories by their mean velocity, the 10th percentile is 0.0027 $\mu$ms$^{-1}$ and

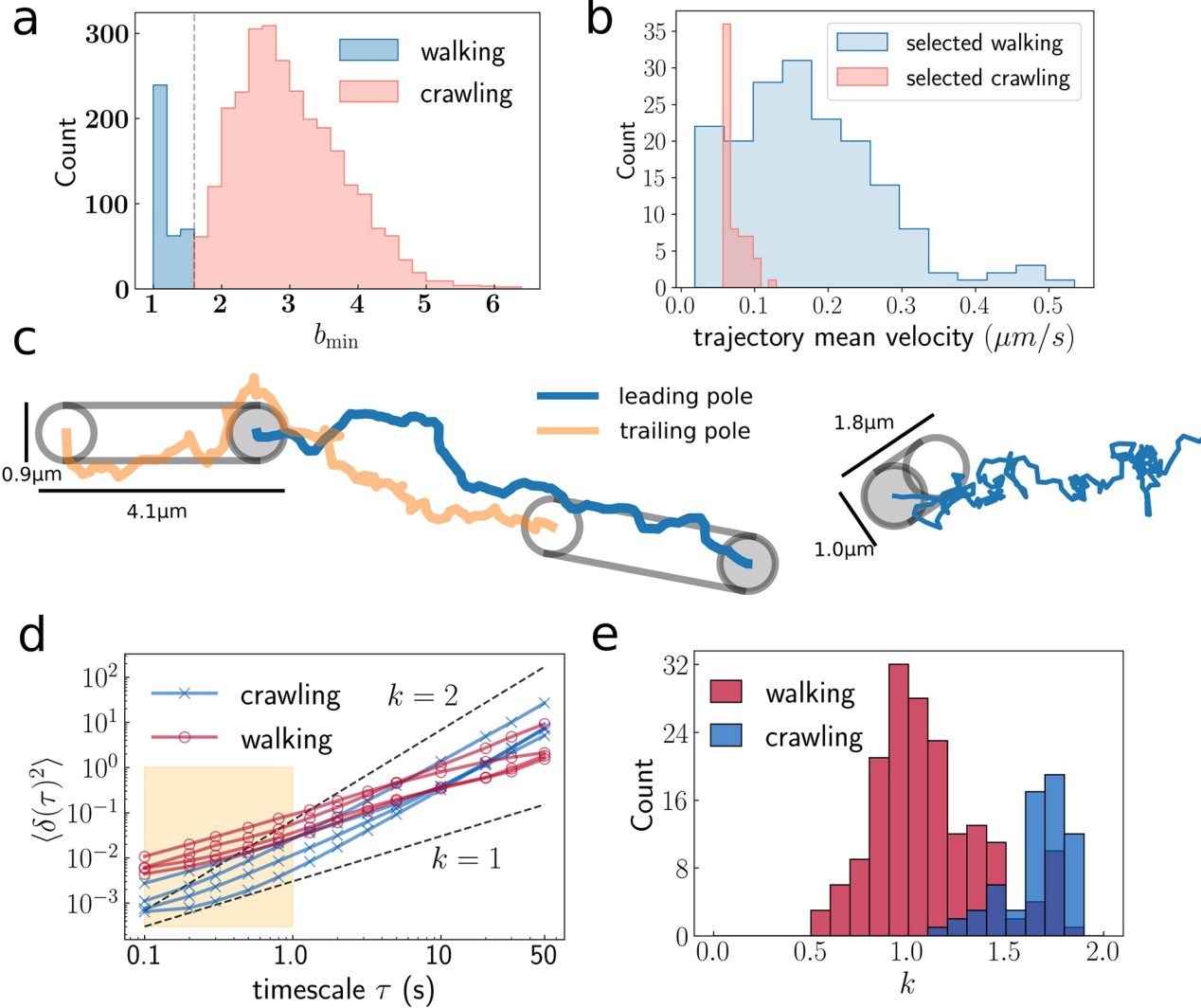

**Fig 3. Classification of trajectories.** (a) Distribution of per-trajectory minimum aspect ratio used as a first step to classify trajectories. (b) Per-trajectory mean velocity distributions for 175 walking trajectories and 63 crawling trajectories what were filtered by hand from 371 potential walking trajectories and 100 crawling trajectories with similar mean velocity. (c) Example trajectories, 100 seconds of a typical crawling trajectory and 220s trajectory of the leading pole of a typical walking trajectory. (d) Sample mean squared displacement curves for selected walking and crawling trajectories. (e) Distribution of $k$ statistic for selected walking and crawling trajectories. $k = 1$ is diffusive motion and $k = 2$ is ballistic motion.

the 90th percentile is 0.075 $\mu$ms$^{-1}$. Surprisingly, the slow crawling trajectories are not typically composed of long waiting periods interspersed by large, rapid displacements as might be expected according to the twitching mechanism. Instead, almost all such trajectories make slow and consistent forward progress. A possible explanation is that these bacteria are tightly adhered to the substrate which strongly affects their TFP retraction speed and another is that these bacteria have many bound TFP which inhibits their motion. On the other hand, approximately linear sub-micrometer displacements, that we associate with the retraction of TFP, are visible in faster trajectories. The measure of the relevant information about the TFP dynamics contained in a given trajectory depends on the relative size of these displacements compared to the spatial resolution, and on the duration of the retraction events compared to the temporal resolution. Therefore, it is more promising to investigate fast trajectories than slow ones, if the goal is extract quantitative information about the TFP processes that generated these trajectories.

According to this argument, we selected a set of crawling trajectories from the high mean–velocity end of the distribution to compare with the computer model. The crawling trajectories are first filtered to remove those with duration less than 100 s and those which displace by less than one hundred 0.12 $\mu$m steps after linearisation. Secondly, only trajectories that have a total end-to-end displacement of more than 3 $\mu$m (approx. one bacterial length) are selected. After the filtering, 100 crawling trajectories with the highest mean velocities are used as a crawling dataset for further analysis. Identical filtering is applied to the walking trajectories, after which there are 160 remaining which we take as our walking dataset.

Given these sets of experimental trajectories, the next section deals with calibrating the model to produce simulated trajectories that have similar statistics. Before developing a set of statistical tools to accomplish this, it is helpful to look at a few example simulated trajectories to see that the model can produce trajectories with statistics that are, at a glance, visually indistinguishable from real twitching trajectories, see Fig 4.

## 4 Parameter inference

With properly processed and classified experimental data, we can run simulations to search for model parameters that result in simulated trajectories with the closest possible resemblance to the chosen experimental data. To do this, we first need to define suitable summary statistics, *i.e.*, a measure for how similar two trajectories are. Since our model has 14 parameters, a full parameter space search is inaccessible, and we restrict the number of parameters that we vary.

We start by fixing several parameters in which we have the highest confidence. Since we assume surface sensing behaviour [6], $1/k_{\text{ret,on}} \rightarrow 0$ is a good model for surface interacting TFP (see also section 2). For unbound TFP, Koch et al. are confident in the estimate $1/k_{\text{ext,on}} = 0.4$ s [5], so we fix this parameter. We also fix $v_{\text{ext}}$ using the measurement of Zhang et al. [28] but keep $v_{\text{ret}}$ as a variable. Varying both $E$ and $f_{\text{stall}}$ is redundant, hence we also fix $f_{\text{stall}} = 100$ pN.

For the remaining parameters, we perform the sensitivity analysis and identify those whose variation does not significantly affect the summary statistics. Subsequently, we select four parameters that significantly affect the behaviour of our model and cannot be extracted accurately from existing experimental data. Finally we estimate these four parameters using approximate Bayesian computation.

### Summary statistics

We define four summary statistics summarized in Table 3, selected so that they reflect short timescale ($<1$ s) structure of the trajectory associated with the TFP retraction dynamics and, as a set, contain as little redundant information as possible.

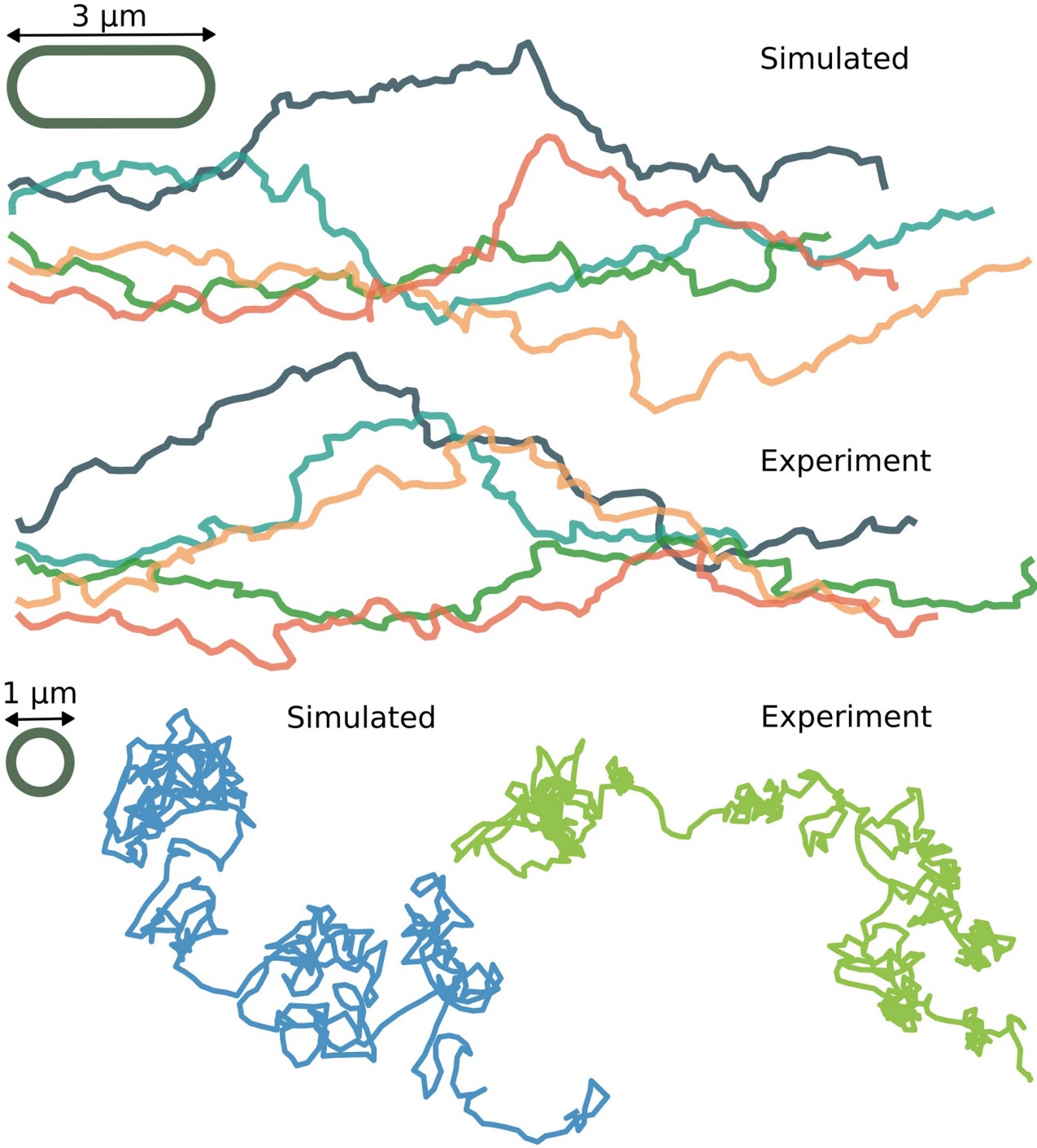

**Fig 4. Example simulated and experimental trajectories.** (Upper) The leading pole of five simulated crawling trajectories and five experimental crawling trajectories are truncated to (330 s) and rotated so that their end-to-end vectors are horizontal. A spherocylinder with length 3 μm and width 1 μm is drawn for scale. (Lower) A typical experimental walking trajectory that was tracked for a total of 870 s and a simulated walking trajectory that is truncated to the first 1000 s. The simulated trajectories are calibrated to match experimental data by the parameter inference method developed in section 4.

**Table 3. Summary statistics.** All statistics are computed on linearised trajectories with $s_i$ the step index. $\boldsymbol{v}$ is the velocity of the leading pole. $\boldsymbol{b}$ is the body axis projected onto the surface plane. $\langle\cdot\rangle$ is an average over linearised steps and over sets of trajectories. A set of simulated trajectories are repeated simulations which differ only in their random seed while experimental trajectories are grouped as in section 3. $n$ is the total number of linearised steps in the set of trajectories. $\hat{q}$ and $\hat{a}$ are the maximum likelihood estimates for a persistent random walk described by $\boldsymbol{v}_{s_i} = \hat{q}\boldsymbol{v}_{s_{i-1}} + \hat{a}\boldsymbol{n}_{s_i}$ where the $\boldsymbol{n}_{s_i}$ are uncorrelated, normally distributed random numbers with unit variance. The estimate of $\hat{a}$ depends on $\hat{q}$. $\hat{q}$ is sensitive to errors in the tracking data. To avoid bias from measurement errors, we discard the top 1% of linearised velocities in the computation of $\hat{q}, \hat{a}$ (see S7 Appendix).

| Symbol | Statistic | Description |
|---|---|---|
| $\langle u \rangle$ | $\langle \lvert \boldsymbol{v} \rvert \rangle$ | Mean speed of leading pole |
| $Var(\theta_d)$ | $Var(\cos^{-1}((\boldsymbol{b} \cdot \boldsymbol{v})/\lvert\boldsymbol{b}\rvert\lvert\boldsymbol{v}\rvert))$ | Variance of deviation angle |
| $\hat{q}$ | $\sum_{s_i} \boldsymbol{v}_{s_i} \cdot \boldsymbol{v}_{s_{i-1}} / \sum_{s_i} \boldsymbol{v}_{s_{i-1}} \cdot \boldsymbol{v}_{s_{i-1}}$ | Persistence of a persistent random walk [37] |
| $\hat{a}$ | $\sqrt{\frac{1}{2(n-1)}\sum_{s_i}\left(\boldsymbol{v}_{s_i} - \hat{q}\boldsymbol{v}_{s_{i-1}}\right)^2}$ | Activity of a persistent random walk [37] |

We compute these statistics using linearised data with $\delta_{\text{step}}$ = 0.12 μm. At this step size, the median step time for crawling data is 1.6 s (walking 0.5 s), which is comparable to the extension and retraction timescales of TFP. At this spacial and temporal resolution we believe there may be some unresolved TFP activity that is relevant to our analysis, however if we reduce the step size our analysis may no longer be robust to measurement noise.

The $\hat{q}, \hat{a}$ statistics measure distinct properties of the shape of the trajectory while the mean velocity $\langle u \rangle$ measures the rate at which the bacteria trace out this shape, hence each of these statistics contain some unique information. The $Var(\theta_d)$ statistic is calculated using the bacteria orientation as well as its trajectory. Therefore we expect that none of the statistics in Table 3 are redundant.

## Deriving a minimal model using sensitivity analysis

Of the 14 parameters in Table 1, we listed 11 with estimates from various experiments. We emphasize that, while these estimates are all obtained for the flagella deficient ΔfliC mutant of *P. aeruginosa*, the experimental procedures and conditions and even the bacterial strain are not consistent. The twitching behaviour of bacteria is observed to be sensitive to temperature, nutrient availability [14] and surface topography [11] among other factors [10, 13, 30]. Behaviour also necessarily varies over the course of surface colonisation [22, 38]. Furthermore some characteristics of *P. aeruginosa* are observed to vary widely between individuals in a single population, in particular $k_{\text{spawn}}$ [5], and the surface adhesion strength [15]. The population variability and sensitivity to the environment of the TFP distribution, the TFP surface dwell time and the motor binding/unbinding rates are unknown.

We deal with these uncertainties by evaluating whether some parameters have a more significant effect on simulated trajectories than others. This is done by computing the sensitivity of our summary statistics to changes in the parameters, within some reasonable upper and lower bounds which are informed by the literature. We then eliminate those parameters that have minimal influence on the behaviour, and in doing so derive a minimal model for the twitching motility of *P. aeruginosa*.

For each summary statistic defined in Section 3, we compute the Sobol global total sensitivity indices [39–41] for our model parameters. As described in S5 Appendix, the sensitivity index for a statistic $f(x_1, x_2, \ldots, x_n)$ and a parameter $x_i$ is a number $S_i^T \in [0, 1]$ which estimates the proportion of the variance of $f$ that is associated with $x_i$. The sensitivity indices we use are described as 'global' because they are computed with respect to a chosen interval in parameter

space, rather than a single point, and as 'total' because they include contributions due to interactions between parameters. For a parameter $x_i$, if any statistic has a large sensitivity index with respect to $x_i$ then we can conclude that our model is sensitive to that parameter on the chosen interval. Conversely if the $S_i^T$ are small for every statistic $f$, we freeze out that parameter and simplify the model while retaining the most significant behaviour.

The results of an 11 parameter sensitivity analysis are shown in Table 4. In total 26624 trajectories were simulated for 2000 seconds at difference parameter values. The bounds of $k_{spawn}$ and $\tau_{dwell}$ are informed by recent experiments [5, 6], and the bounds for $k_{ret,off}$, $k_{ret,on}$, $k_{ext,off}$, $k_{ext,on}$ are approximately the 95% confidence intervals given by Koch et. al. [5]. The remaining parameters are varied by at least an order of magnitude.

We indeed observe that several parameters score low in the sensitivity test. Our model is insensitive to $k_{ext,on}$, which affects the delay between subsequent extensions, since in our model TFP predominantly perform a single extension/retraction cycle. Likewise, the model is insensitive to $k_{ext,off}$ because most TFP fully retract before the motor can unbind. The parameters $k_{resample}$ and $L_p$ affect the extent to which pili flex in order to make contact with the surface. Persistence length estimates vary enormously [4, 27]. The minimum persistence length considered here is 1 μm as this bound appears to be consistent with live imaging of TFP [4, 5], although smaller measurements have been reported [27].

As discussed in section 2, the stall force $f_{stall}$, plays a limited role in this model because we do not include friction or viscous forces explicitly and instead allow these to enter the model indirectly in the TFP retraction speed. Although the stall force $f_{stall}$ and the elastic modulus $E$ could in principle still play a role in the dynamics by determining the maximum extension that a pilus can support relative to its equilibrium length before the motor is stalled, the simulated trajectories turn out not to be sensitive to these two parameters, as demonstrated in Table 4.

On the basis of this sensitivity analysis, we fix the values of $k_{ext,on}$, $k_{resample}$, $k_{ret,off}$, $L_p$, $E$ and $f_{stall}$ to their best estimates. The parameter $k_{ext,off}$ is important, since it determines the typical extension length of TFP. We simplify the model by fixing this parameter to the estimate of Koch et al. [5] on the basis that it may be an intrinsic property of the TFP motor complex and associated proteins, although this as yet to be demonstrated. The remaining four parameters are $\tau_{dwell}$, $\kappa$, $v_{ret}$ and $k_{spawn}$ constitute a minimal model of twitching and represent the biologically meaningful information that we aim to extract from tracking data using approximate Bayesian computation.

**Table 4. Parameter sensitivity.** The Sobol total effect indices for the parameters in Table 1, $v_{ext}$ and $k_{ret,on}$. Sensitivity indices are calculated independently for each summary statistic. Large sensitivity indices are highlighted.

| parameter | min | max | $\langle u \rangle$ | $Var(\theta_d)$ | $\hat{q}$ | $\hat{a}$ | max $S_i^T$ |
|---|---|---|---|---|---|---|---|
| $k_{ext,off}$ | 0.2 | 1 | 0.4407 | 0.0302 | 0.3947 | 0.0085 | 0.4407 |
| $\tau_{dwell}$ | 0.5 | 3 | 0.1161 | 0.0299 | 0.2662 | 0.0064 | 0.2662 |
| $\kappa$ | 1 | 15 | 0.0780 | 0.8693 | 0.2905 | 0.0093 | 0.8693 |
| $v_{ret}$ | 2.5 | 250 | 0.3089 | 0.0487 | 0.2815 | 0.9747 | 0.9747 |
| $k_{spawn}$ | 0.1 | 8 | 0.4167 | 0.0439 | 0.2079 | 0.0077 | 0.4167 |
| $k_{ext,on}$ | 0.2 | 0.5 | 0.0116 | 0.0067 | 0.0492 | 0.0017 | 0.0492 |
| $k_{resample}$ | 0.5 | 10 | 0.0092 | 0.0074 | 0.0482 | 0.0020 | 0.0482 |
| $k_{ret,off}$ | 0.05 | 0.2 | 0.0082 | 0.0063 | 0.0459 | 0.0019 | 0.0459 |
| $L_p$ | 1 | 10 | 0.0073 | 0.0062 | 0.0429 | 0.0018 | 0.0429 |
| $E$ | 1000 | 20000 | 0.0088 | 0.0079 | 0.0576 | 0.0049 | 0.0576 |
| $f_{stall}$ | 20 | 200 | 0.0140 | 0.0631 | 0.0808 | 0.0151 | 0.0808 |

## Approximate bayesian computation

For given tracking data $x$, and model parameters $\theta$, parameter inference is computing the conditional probability distribution $P(\theta|x)$, which is usually called the *posterior*. According to Bayes theorem, $P(\theta|x) \propto P(x|\theta)\pi(\theta)$, where $P(x|\theta)$ is called the *likelihood* and $\pi(\theta)$ the *prior*. Approximate Bayesian computation (ABC) [26] is a method for numerically evaluating the posterior using appropriate summary statistics $s$. We choose a reference data set $x'$ and evaluate the summary statistics for this reference data, $s'$. The reference data set is an experimental trajectory or a set of experimental trajectories, but for purpose of validating the method it can also be simulated trajectories. The next step is to run simulations for sets of parameters sampled from the prior. For each simulation, the obtained summary statistics $s$ is compared to the reference $s'$ by Euclidean metric: $\rho = ||s - s'||$. The straightforward rejection–ABC approach is to evaluate a large number, $N$, samples and accept a small fraction, $M$, of the best samples. If the scores $\rho_i$ are ordered from smallest to largest, the a threshold is defined as $\epsilon = \rho_M$, and $x_{i \leq M}$ are labeled as accepted. These accepted trajectories are interpreted as samples of the posterior distribution [26, 42]. The accepted simulations are the ones that each, to some approximation, relate well to the experimental data. In this section we analyse the properties of these accepted simulations to determine what are the consistent properties of these simulations.

We follow [43] and re-scale the summary statistics by their respective standard deviations across all $N$ samples before comparing them. Another optimisation we use is to weight samples using an Epanechnikov kernel [26] with the width $\epsilon$. Parameter inference methods are applied separately to crawling and walking data.

## 5 Results

### Parameter inference for crawling trajectories

We choose a uniform prior distribution for the four parameters with bounds $\tau_{\text{dwell}} \in [0.05, 3.0]$ s, $\kappa \in [1.0, 15.0]$, $v_{\text{ret}} \in [0.01, 1.0]$ and $k_{\text{spawn}} \in [0.1, 8.0]$ s$^{-1}$. These bounds include a wide range of physically reasonable parameter values as seen in twitching experiments. Approximate Bayesian computation is performed by evaluating $N = 10000$ parameter samples and accepting $M = 50$ samples. Each sample is a set of four parameters obtained from the prior distribution, which we evaluate by simulating trajectories with that parameter set and then computing summary statistics on the resulting trajectories. Trajectories are simulated for 2000 seconds each and up to 10 trajectories are simulated for each parameter sample, however we stop simulating new trajectories once the accumulated trajectory data for that parameter sample reaches a total contour length of 120 μm (1000 linear steps of 0.12 μm).

The results of 4 dimensional approximate Bayesian computation are interpreted by plotting the two–dimensional cardinal projections of the accepted samples in the upper–right triangle of a 4x4 grid. In the lower triangle, we plot projections of the posterior distribution smoothed using weighted kernel density estimation [44]. On the diagonal axes we plot weighted histograms for each parameter (Fig 5). Parameter estimates are obtained by smoothing these weighted histograms using weighted kernel density estimation to obtain a probability distribution for each parameter. The choice of kernel bandwidth is important for reliable parameter estimation so we use cross-validated least squares bandwidth selection [44, 45] to choose an appropriate bandwidth. This approach sidesteps the difficulties in bandwidth selection for multivariate data [46]. The maximum likelihood estimates and 90% confidence intervals for these probability distributions are $\tau_{\text{dwell}} = 2.11[0.93, 2.86]$ s, $\kappa = 2.46[1.83, 3.63]$, $v_{\text{ret}} = 0.17$ [0.11, 0.23] μms$^{-1}$, $k_{\text{spawn}} = 5.55[1.98, 7.49]$ s$^{-1}$. The low TFP retraction speed predicted here is broadly consistent with low retraction speed measured for *P. aeruginosa* bacteria in very

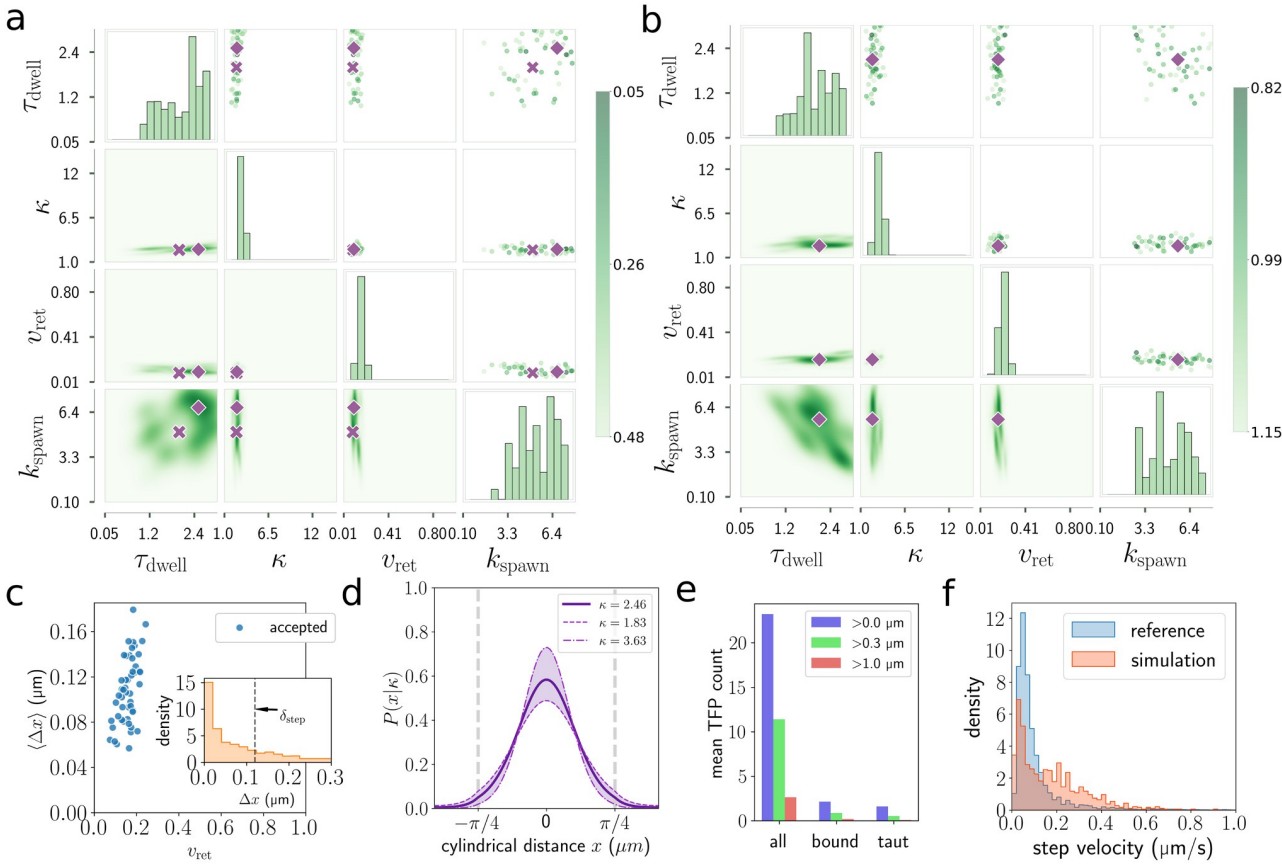

**Fig 5. Parameter inference for crawling trajectories.** (a,b) Cardinal projections of the 4d approximate posterior distribution obtained by accepting 50/10000 samples. The reference data for (a) is simulated data and for (b) it is a subset of the tracking data. Weighted histograms of the accepted parameter values are plotted on the diagonal axes. Crosses mark the parameters of the simulated reference data while Diamonds show the maximum likelihood estimates. Color indicates the sample score, $\rho$. (c) Blue circles are the mean per-TFP displacements, $\langle \Delta x \rangle$, of the accepted samples. The inset figure is the $\Delta x$ distribution for the best scoring simulation. (d) Probability density function used for generating TFP on the body surface. The distribution is extended onto the cylindrical portion of the spherocylinder as described in section 2. Shaded region is the 90% confidence interval in the ABC estimate of the $\kappa$ parameter. (e) Counting TFP using the best scoring simulation. the mean numbers of total TFP, bound TFP and taut TFP are shown as well as those counts in the cases that only TFP longer than 0.3 μm and 1.0 μm are visible to the observer. (f) Linearised velocity distributions for experimental tracking data (blue) and the best scoring simulated data (orange).

similar conditions as compared to our tracking dataset [28]. The wide distribution of TFP on the leading pole appears consistent with recent imaging methods [5, 6], although quantitative experimental measurements of the distribution are still unavailable. Note, we do not regard the upper bounds of the $\tau_{\text{dwell}}$ and $k_{\text{spawn}}$ parameters as meaningful in this case since they are approximately the upper bounds of the search space.

**Validation on simulated data.** With these estimates in hand, we selected the following parameters ($\tau_{\text{dwell}}$, $\kappa$, $v_{\text{ret}}$, $k_{\text{spawn}}$) = (2.0 s, 2.5, 0.09 μms$^{-1}$, 5.0 s$^{-1}$) to use while validating the parameter inference methods. $v_{\text{ret}}$ is the value measured by Zhang et al. [28] and the other values are suggested by the initial parameter inference. We validate the method by attempting to predict these parameters from simulated data. Simulated trajectories with these parameters have the following statistics ($\langle u \rangle$, $Var(\theta_d)$, $\hat{q}$, $\hat{a}$, ) = (0.055 μms$^{-1}$, 1.03, 0.42, 0.10), which we now take as the reference statistics $\boldsymbol{s}'$.

The parameter estimates that we recover are $\tau_{\text{dwell}}$ = 2.51[1.05, 2.90] s, $\kappa$ = 2.56[1.91, 3.21], $v_{\text{ret}}$ = 0.10[0.06, 0.17] μms$^{-1}$, $k_{\text{spawn}}$ = 6.72[2.76, 7.64] s$^{-1}$. Notice sharp estimates for $\kappa$, $v_{\text{ret}}$ and

large uncertainties in the estimates for the $\tau_{\text{dwell}}$ and $k_{\text{spawn}}$ parameters even when using simulated reference data (Fig 5A). The usefulness of this validation method is that we know that the difficulties in recovering simulation parameters from simulated data are due either to the limitations of the trajectory information or shortcomings in our parameter inference methods, that is, the twitching model itself is not the problem. We have noted that significant information about the TFP activity could be lost by coarse graining at a length of 0.12 μm. Since the simulated data is noise free, we investigated whether parameter inference would be more accurate by using the original 0.1 *s* resolution simulated data without any processing. However, the 90% confidence interval for $k_{\text{spawn}}$ is just as broad as before.

Another possibility is that the trajectory data does contain sufficient information to distinguish between trajectories with varying $k_{\text{spawn}}$, $\tau_{\text{dwell}}$ but the summary statistics are insensitive to this information. Therefore we searched for statistics that could outperform the the ones we use here. The mean squared displacements $\langle \delta(\tau)^2 \rangle$ of the trajectory, where the mean is computed over all pairs of measurements separated by the timescale $\tau$, has been used to characterise the persistence of biological trajectories [47, 48]. Such organisms commonly exhibit short timescale stochastic behaviour, persistence at medium timescale, and diffusive behaviour on long timescales. Our $\sim 30$ min experiments are too short to observe long timescale diffusion and $\hat{q}$ is a measure of the short length-scale persistence. To analyse the medium time-scale persistence we write $\langle \delta(\tau)^2 \rangle \sim \tau^k$ and compute $k$. We cut off the short timescale behaviour for this calculation using a threshold $\tau \geq 2$ s. The experimental crawling trajectories are nearly ballistic on this timescale with a mean $k$ of 1.86. However, including this statistic does not improve the parameter inference on simulated data, in fact, using the standard statistics in Table 3 we find the 50 accepted simulations have a mean $k$ of 1.78, similar to the reference data even when $k$ is not used as a statistic.

Noting that the retraction of TFP leads to a series of approximately linear displacements on short timescales, we used piecewise-linear regression to extract the length and time distributions of these linear features. Using statistics from these distributions also did not improve parameter inference on simulated data. Afterwards we consider statistics which are not attainable from experimental data, such as the mean numbers of taut $\langle N_{\text{taut}} \rangle$ and bound $\langle N_{\text{bound}} \rangle$ TFP and the mean per-pilus displacement $\langle \Delta x \rangle$. The latter is computed by summing the small displacements due to individual retraction events over the lifetimes of each TFP. Even with this information, we did not find any combination of statistics that perform significantly better than those in Table 3.

**Per-TFP displacement as an invariant quantity.**  Since the uncertainties in the estimates of $k_{\text{spawn}}$, $\tau_{\text{dwell}}$ were consistently large, we turn to an alternate line of questioning. What is it about the accepted trajectories that are so similar, even while $k_{\text{spawn}}$, $\tau_{\text{dwell}}$ are varying? By construction, the accepted samples already have mean velocities $\langle u \rangle$, deviation angle distributions $Var(\theta_d)$, and shapes $(\hat{q}, \hat{a})$ that are similar to the reference data and to each other. We continue by comparing mean taut and bound TFP numbers. In this simplified TFP model, only taut TFP directly contribute to the dynamics so we focus on the $\langle N_{\text{taut}} \rangle$ statistic, but similar results are found for $\langle N_{\text{bound}} \rangle$. Based on prior work, we expect $\langle N_{\text{taut}} \rangle$ to have a significant effect on the characteristics of the trajectory [24]. By computing the Pearson correlation coefficient for the accepted samples, we find that $\langle N_{\text{taut}} \rangle$ depends heavily on $k_{\text{spawn}}$ (coefficient $r = 0.75$) and $\tau_{\text{dwell}}$ ($r = 0.74$) suggesting that the degeneracy in these parameters contributes to the difficulty of predicting each of them.

Next, we investigate how trajectories can have varying $\langle N_{\text{taut}} \rangle$ but be otherwise similar according to the statistic vector **s**. We compute the ratio of standard deviations of accepted samples to that of all samples for $\langle N_{\text{taut}} \rangle$ to be 0.69/0.86 = 0.80, the same calculation for the mean per-TFP displacement $\langle \Delta x \rangle$ is 0.031/0.123 = 0.256. The result indicates that trajectories

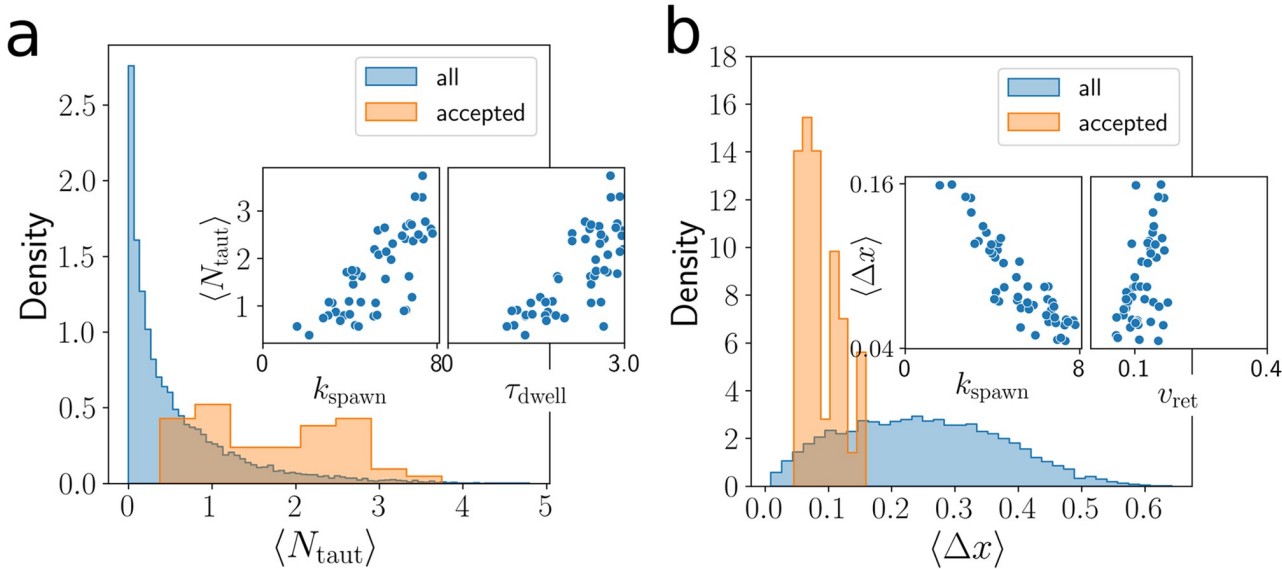

**Fig 6. Properties of best-matching trajectories.** (a) The distribution of $\langle N_{\text{taut}} \rangle$ for all ABC samples and for the $N = 50$ accepted samples using the score function $||s - s'||$ where $s'$ are a vector of statistics computed from simulated trajectory data. Insets show the correlation between $k_{\text{spawn}}$, $\tau_{\text{dwell}}$ and $\langle N_{\text{taut}} \rangle$ for the accepted samples. (b) Distribution of $\langle \Delta x \rangle$ for all ABC samples and for the accepted samples. Insets show the correlation between $k_{\text{spawn}}$, $v_{\text{ret}}$ and $\langle \Delta x \rangle$ for the accepted samples.

which have similar statistics $s$ are similar in $\langle \Delta x \rangle$ but not necessarily in $\langle N_{\text{taut}} \rangle$ (Fig 6). We find that $\langle \Delta x \rangle$ is anticorrelated with $k_{\text{spawn}}$ ($r = -0.89$), and correlated with $v_{\text{ret}}$ ($r = 0.51$). The correlation between $k_{\text{spawn}}$ and $v_{\text{ret}}$ is $r = -0.28$ indicating that these two parameters have an inverse relationship that makes parameter inference challenging.

### Parameter inference for walking trajectories

We now investigate whether it is possible to use the walking motility state of *P. aeruginosa* to expand and cross-validate our parameter inference. Approximate Bayesian computation is performed again using a uniform prior with bounds $\tau_{\text{dwell}} \in [0.05, 3.0]$ s, $\kappa \in [1.0, 15.0]$, $v_{\text{ret}} \in [0.01, 1.0]$ and $k_{\text{spawn}} \in [0.1, 8.0]$ s$^{-1}$, and again a sampling size of $N = 10000$ but this time we initialise the simulation in the walking state and use a hard repulsive surface. The method returns estimates for the parameters, $\tau_{\text{dwell}} = 1.46[0.92, 2.27]$, $\kappa = 1.76[1.22, 2.34]$, $v_{\text{ret}} = 0.50$ $[0.29, 0.83]$ μms$^{-1}$, $k_{\text{spawn}} = 4.18[2.50, 7.53]$ s$^{-1}$ (Fig 7B). Strikingly, the estimate of the TFP retraction speed is much larger than for crawling trajectories. The value of $0.5$ μms$^{-1}$ is consistent with the value measured by Skerker & Berg. In the walking state, *P. aeruginosa* have a much smaller contact area with the surface. One interpretation of the differing retraction speed estimates is that bacteria in the walking state are relatively unimpeded by surface friction forces, while crawling bacteria have stronger surface adhesion which corresponds to slower retraction speeds [29]. Regarding the other parameters, the estimate for the dwell time is roughly consistent with a recent experimental measurement [6]. The TFP distribution width and spawn rate estimates are similar to those for crawling bacteria.

We again generated simulated reference data and used it to test the effectiveness of our parameter inference method. In this case we will use the crawling reference parameters initially but set $v_{\text{ret}} = 0.5$ μms$^{-1}$. The simulated parameters used are $(\tau_{\text{dwell}}, \kappa, v_{\text{ret}}, k_{\text{spawn}}) = (2.0$ s, $2.5$, $0.5$ μms$^{-1}$, $5.0$ s$^{-1}$), the recovered parameters are $\tau_{\text{dwell}} = 2.25$, $[1.52, 2.89]$ s, $\kappa = 2.33[1.53, 9.74]$, $v_{\text{ret}} = 0.43[0.1, 0.89]$ μms$^{-1}$, $k_{\text{spawn}} = 4.21[2.26, 7.36]$ s$^{-1}$.

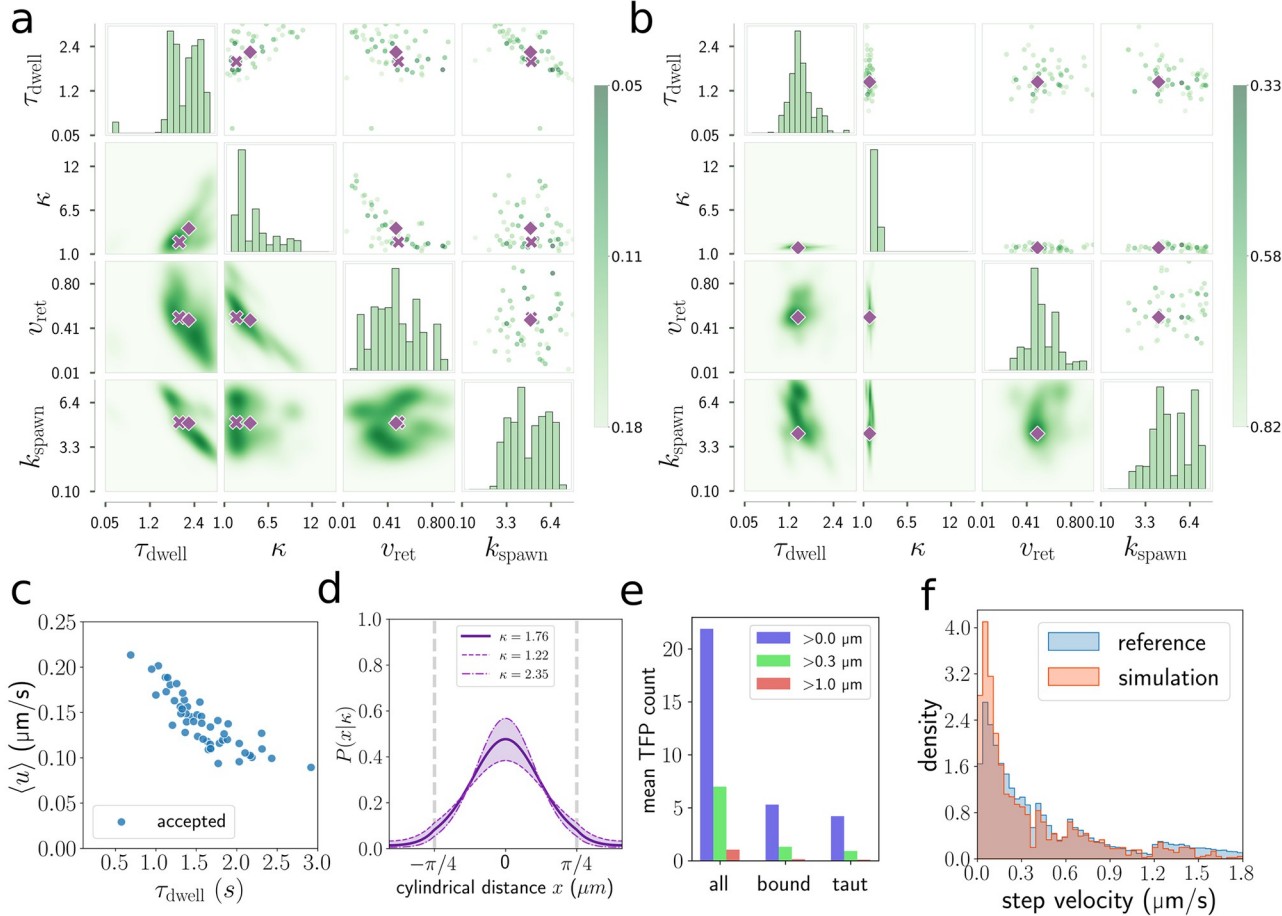

**Fig 7. Parameter inference for walking trajectories.** (a,b) Cardinal projections of the 4d approximate posterior distribution obtained by accepting 50 parameter samples from 10000 walking simulations. The reference data for (a) is simulated data with parameters ($\tau_{\text{dwell}}$, $\kappa$, $v_{\text{ret}}$, $k_{\text{spawn}}$) = (2.0 s, 2.5, 0.5 $\mu\text{ms}^{-1}$, 5.0 $\text{s}^{-1}$) and for (b) it is walking trajectories identified from experimental tracking data. (c) Blue circles are the mean velocity, $\langle u \rangle$, of the accepted samples. For (d,e,f) see the caption in Fig 5.

**TFP distribution width.** The TFP distribution width parameter $\kappa$ has sharp estimates using both walking and crawling data, they are $\kappa = 1.76$ and $\kappa = 2.56$. These values correspond to broad TFP distributions, see Fig 2 for a visual representation. For walking bacteria, TFP at the tails of the distribution rarely interact with the surface, hence we may find it difficult to distinguish TFP distributions with very low $\kappa$ in this state. Even so, the 90% confidence intervals for this parameter overlap between walking and crawling trajectories, suggesting that the TFP distributions for walking and crawling bacteria may be similar.

**Dwell time.** According to the simulated parameter inference, the $\tau_{\text{dwell}}$ parameter can be estimated from walking trajectories whereas it is more challenging to do so for crawling trajectories. Our estimate for walking bacteria is $\tau_{\text{dwell}} = 1.46$ s, consistent with that of the singular study that directly measures surface binding time, finding it to be 1.0 s with 95% confidence interval [0.59, 1.80] [6].

An inspection of the correlation coefficients for the ABC accepted samples indicates why estimates from walking trajectories are more specific. The parameter $\tau_{\text{dwell}}$ and mean velocity $\langle u \rangle$ are strongly anticorrelated with coefficient $r = -0.84$ (Fig 7C), compared to the same correlation for crawling trajectories, $r = -0.50$. We argue that in the walking state, TFP binding to

the surface for an excess time significantly inhibits motility and that TFP pull in opposing directions more often—two effects that are diminished in the crawling state. Our results indicate that it may be possible to use the the different motility states of *P. aeruginosa* to infer different TFP properties from tracking data.

**Counting TFP.** Figs 5E and 7E show the mean numbers of TFP for the best scoring walking and crawling simulations. For both walking and crawling data there are large uncertainties in our estimates for the TFP spawn rate $k_{spawn}$, however the lower bounds of the confidence intervals, 1.98 TFP/s and 2.26 TFP/s are similar. A singular experiment for *P. aeruginosa* in an optical trap reports that this value varies significantly among individual bacteria but is rarely more than 0.5 TFP/s [5]. We first point out that there is no reason to expect bacteria in different environments to have the same TFP characteristics, however by comparing spawn rate and mean numbers of TFP with other TFP imaging experiments [4, 6], we find that our model predicts surprisingly large numbers of TFP. However, we must take into to account the optical resolution of these experiments. The optical trap experiment cannot detect TFP less than 0.3 μm while the other recent live cell imaging experiment cannot detect TFP less than around 1.0 μm [6]. We therefore also count TFP using these thresholds. For the best scoring crawling simulation an average of 11.4/23.2 TFP are visible at the 0.3 μm threshold and this drops to 2.6/23.2 for the 1.0 μm threshold. For the best scoring walking simulation, these numbers are 7.0/21.9 and 1.0/21.9 for the 0.3 μm and 1.0 μm thresholds respectively. Our finding is that simulations which match well to the experimental data often have large numbers of TFP but many are too short to interact with the surface or be visible in experiments.

The typical numbers of bound TFP for this pair of crawling and walking simulations are 2.1 and 5.3 respectively and the numbers of taut TFP are 1.6 and 4.2. We cannot verify predictions of the mean numbers of bound and taut TFP from tracking data alone so we conclude our analysis with several observations. Firstly, walking bacteria have their TFP growth oriented towards the surface so it is natural that they have more surface interacting TFP than crawling bacteria. Secondly, we computed the fraction of TFP that interact with the surface before they fully retract and are dissolved, for crawling simulations (Fig 5A). It is 0.18 for the accepted samples and 0.11 for all samples, meaning that most TFP do not contribute to the bacteria motility at all. This conclusion is supported by a recent experimental study [29]. We are interested to know whether *P. aeruginosa* has a mechanism to make more efficient use of its TFP, for example should the bacteria preferentially construct TFP on the surface-facing side of its body, they would obtain similar motility but with greater TFP efficiency. Finally, a previous two dimensional crawling *P. aeruginosa* model shows that bacteria with approximately 2 to 3 surface bound TFP have twitching-like trajectories with rich dynamics [24]. The same is true of our model.

## Walking ↔ Crawling Transitions

TFP are recognised as an important component in the surface adhesion of *P. aeruginosa* [15]. However, surface adhesion due to TFP alone does not explain the crawling state. In our simulations, model bacteria with a purely repulsive body–surface interaction immediately transition to a walking state due to an unbalanced vertical component in the forces generated by TFP retraction (Fig 1C). We measure transition rates by initializing 100 independent trajectories in the crawling state and simulating for $t_{max}$ = 200 seconds. We record the transition time $\tau_w$ at which the surface contact of the trailing pole is broken. If no transition occurred then we simply set $\tau_w = t_{max}$. Fig 8C shows that the transition to walking is controlled by the surface interaction strength $\epsilon$ (S1 Appendix). We are confident based on the surface–projected cell body aspect ratio that at least 80% of trajectories in our tracking data remain in the crawling

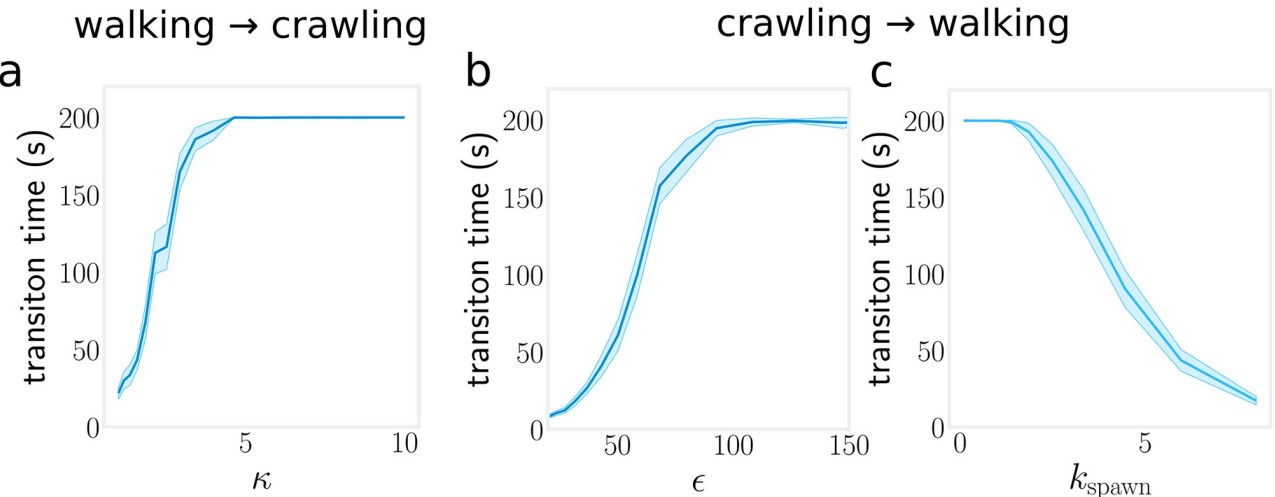

**Fig 8. Walking to crawling transitions.** Starting with a base set of parameters ($\tau_{\text{dwell}}$, $\kappa$, $v_{\text{ret}}$, $k_{\text{spawn}}$) = (1.0 s, 2.5, 0.5, 5.0 s$^{-1}$), (a) the rate of walking → crawling transitions depends on the pili distribution width parameter $\kappa$. (b) Crawling → walking transitions occur by reducing the surface attraction parameter $\epsilon$. (c) Crawling → walking transition rate vs. $k_{\text{spawn}}$ for $\epsilon$ = 50. Shaded region is the 95% confidence interval obtained by bootstrapping.

orientation and do not transition during the 30 minute experiment. Inspecting the remaining trajectories, confirms that transitions between walking and crawling states are rare. We believe that the prevalence and stability of the crawling state is best explained by a short range TFP–independent adhesive interaction between the bacteria and the surface. *P. aeruginosa* is known to express various non-pili related adhesive proteins that could mediate this adhesive behaviour [49, 50]. This bacteria also excretes the exopolysaccharide Psl which is known to have a role in its surface adhesion [10, 51].

*P. aeruginosa* has been observed to transition to the walking state prior to detaching from the surface [7]. Our results indicate that *P. aeruginosa* that are actively twitching will transition into the walking state naturally following a reduction in surface adhesion strength. In addition, Fig 8D shows that TFP can cooperate to overcome the surface attraction, suggesting an alternative strategy in which the bacteria may upregulate TFP activity to drive the transition. This implies that *P. aeruginosa* may also transition to the walking state stochastically by randomly attaining a configuration in which multiple TFP are retracting simultaneously.

For model bacteria initialized in the walking state, the walking to crawling transition is detected when the trailing pole makes contact with the surface. In our model, this transition depends on a specific configuration of TFP. The transition takes place when a pilus that is extended from the cylindrical portion of the bacteria's spherocylindrical body retracts completely. The frequency of such TFP and therefore the transition rate depends primarily on the $\kappa$ parameter (see Fig 8A). *P. aeruginosa* has been observed to reverse its leading pole on encountering an obstacle [13] and appears to lose its polarisation entirely during irreversible surface attachment [22]. Likewise we suggest that *P. aeruginosa* may adapt its TFP distribution to control its surface orientation.

## Discussion

Surface motility is central to the function and proliferation of many micro-organisms. With this in mind it is not surprising that motile behaviours are highly sensitive to environmental conditions, making it difficult to quantitatively compare the various experiments that have

been reported to date. Progress is further hampered by the multi-scale nature of the problem ranging from molecular to micrometer scales. This is reflected in models with a large number of assumptions and parameters—a generic difficulty in modelling complex systems.

In this work we construct a physical model of twitching motility and perform simulations connecting the microscopic properties of the TFP machinery to statistical properties of twitching trajectories. The model is informed both by recent experimental observations of TFP extension and retraction behaviours [5] and TFP–surface interactions [6]. Even so, several simplifying assumptions are made, which are summarised in section 2. Experimental data is used to inform model parameters where appropriate, however, we avoid relying heavily on experimental parameters that have already been reported inconsistently between experiments, and those that have not yet been duplicated. Instead, a systematic strategy supported by sensitivity analysis is used to eliminate weakly influential parameters until we obtain a minimal model that reproduces the complex dynamics observed in twitching bacteria. Constructing a model with a minimal number of variable parameters is not only convenience, doing so also helps us to understand the essential components of twitching behaviour.

This analysis yields an estimate for the retraction speed of TFP that are actively pulling the cell: $v_{\mathrm{ret}} \approx 0.17 \; \mathrm{\mu m s}^{-1}$, significantly less than the number reported by Skerker & Berg. The experiment that uses the most similar procedures as compared to our tracking data reports a retraction speed of $0.09 \; \mathrm{\mu m s}^{-1}$ for loaded, and $0.17 \; \mathrm{\mu m s}^{-1}$ for unloaded TFP [28]. Our model also reproduces qualitative features of experimental trajectories such as velocity profiles and directional persistence of walking and crawling states. In addition, we demonstrate mechanisms by which *P. aeruginosa* may control its surface orientation and hence its directional persistence, a property that is known to be crucial for adapting to different environments [14].

Quantitative comparison of the simulated and experimental trajectories relies on the choice of a suitable summary statistics. The statistics commonly used to study biological trajectories, which are derived from the mean squared displacement and velocity auto-correlation functions, are not sensitive to the short length and time-scale properties of twitching motion. We considered various statistics and chose four that are sufficiently sensitive to long and short timescale features of the trajectories (see Table 3). Choosing optimal summary statistics or otherwise constructing an optimal similarity metric for biological tracking data is itself a challenging problem, and an important area of research [37, 48].

The consistency check, *i.e.*, testing our parameter inference on simulated data, reveals which model parameters are likely to be estimated with precision from the tracking data, and which are challenging to estimate. In fact, due to the nature of the trajectories obtained from walking and crawling motility types, some parameters can be estimated accurately for crawling bacteria but not for walking bacteria and vice-versa, opening the door for parameter inference methods that use both motility types to obtain more information about TFP behaviour than is possible otherwise. Testing on simulated data indicates that the width of the TFP distribution can be estimated with precision for both motility types, and in fact, independent estimates for $\kappa$ for crawling and walking data overlap. Such estimates are important because the TFP distribution has not yet been reported from experiments. Recent advances in live imaging of TFP during twitching [5, 6], indicate that it may soon be possible to verify our predictions experimentally.

The accuracy of our parameter inference methods are limited by a number of factors: the accuracy of our model, the density of parameter samples used for approximate Bayesian computation, the quality of the summary statistics, and the pre-processing applied to the trajectory data. Regarding the latter, for the purpose of our analysis, we coarse-grain both experimental and simulated data to obtain comparable noise-free, trajectories. The median step time after coarse graining at a fixed length of $0.12 \; \mathrm{\mu m}$ is $1.6 \; \mathrm{s}$ for crawling bacteria, similar to the typical

extension, retraction and surface association timescales of TFP [6]. In simulations, we consistently observe that trajectories are the result of cooperative action of many TFP, each contributing a very small displacement, suggesting that the twitching motility of *P. aeruginosa* is governed by physical processes on length- and timescales below this resolution. Therefore, in the available tracking data, the relevant mechanisms might not be resolved directly but only through their integrated effect. We would like to stress that the model developed and analyzed in this work is not intended to be a universal model for twitching motility of *P. aeruginosa*. Rather, the parameters, and perhaps even model assumptions will likely vary with bacteria strains and environmental conditions. The procedure presented here would thus need to be repeated when applying our methods to sufficiently different experiments. It would be exciting to understand to what extent the twitching mechanism varies with external conditions, and we hope that our work will motivate and inform the design of new tracking experiments and filament-resolved imaging with improved spatio-temporal resolution.

The methodology developed in this work is not strictly limited to describe the twitching motility of *P. aeruginosa*. It can be straightforwardly extended to study twitching in different conditions, on different surfaces and for other micro-organisms with various shapes and TFP characteristics. Our model and the simulation codes available online provide a new tool for *in silico* exploration of aspects of twitching motility in unfamiliar environments such as those with micron scale topographic features [11].

## Supporting information

**S1 Appendix. Surface Interaction.** The interactions between the bacteria and the environment surface in our model.
(PDF)

**S2 Appendix. Axis–angle representation.** Details of the calculations for optimising the position and orientation of a rigid body.
(PDF)

**S3 Appendix. Worm-like chain generator.** Details of the calculations for a sampling TFP configurations with respect to their Boltzmann weight.
(PDF)

**S4 Appendix. Comparing retraction speed and $\alpha$ parameters.**
(PDF)

**S5 Appendix. Sobol sensitivity analysis.** Introduction to the sensitivity analysis method used in this work.
(PDF)

**S6 Appendix. Surface sensing.** Analysis showing that it is difficult for our methods to differentiate between different types of surface sensing behaviour that have been suggested for *P. aeruginosa*.
(PDF)

**S7 Appendix. Robustness of random walk statistics.** The maximum likelihood estimators for random walk parameters that we use are sensitive to extremely large fluctuations in velocity. The problem is solved by setting a velocity threshold in our calculations.
(PDF)

**S8 Appendix. Surface adaptation and Psl trails *P. aeruginosa* is known to modify its local micro-environment by excreting EPS.** The extracellular polymers may in turn modify the

motility characteristics of the bacteria. The experiments we study are relatively short and we do not see significant variations in motility that could be attributed to surface deposits of EPS. (PDF)

**S9 Appendix. Aspect ratio profiles.** The aspect ratio of the surface projected bodies of the bacteria are used to identify crawling and walking behaviour. The robustness of this method is discussed and examples of aspect ratio time series are shown. (PDF)

**S1 Video. Example crawling simulation.** 100s of simulation at 10× speed. Parameters ($\tau_{dwell}$, $\kappa$, $k_{spawn}$) = (1.0 s, 2.5, 5.0 s$^{-1}$). (MP4)

**S2 Video. Example walking simulation.** 100s of simulation at 10× speed. Parameters ($\tau_{dwell}$, $\kappa$, $k_{spawn}$) = (1.0 s, 2.5, 5.0 s$^{-1}$). (MP4)

## Acknowledgments

We thank Fan Jin and Lei Ni for providing the tracking data, and Rastko Sknepnek, Bryn Jones, James Farrell, Gerard C. Wong and Calvin Lee for discussions and their valuable input. D.L.B. acknowledges the hospitality, support and resources of Rastko Sknepnek and the University of Dundee while displaced due to the COVID-19 pandemic.

## Author Contributions

**Conceptualization:** Daniel L. Barton, Jure Dobnikar.

**Data curation:** Daniel L. Barton.

**Formal analysis:** Daniel L. Barton.

**Funding acquisition:** Jure Dobnikar.

**Investigation:** Daniel L. Barton.

**Methodology:** Daniel L. Barton, Yow-Ren Chang, William Ducker, Jure Dobnikar.

**Project administration:** Jure Dobnikar.

**Software:** Daniel L. Barton.

**Supervision:** Jure Dobnikar.

**Visualization:** Daniel L. Barton.

**Writing – original draft:** Daniel L. Barton.

**Writing – review & editing:** Yow-Ren Chang, William Ducker, Jure Dobnikar.

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
