## [Decision Letter · Decision Letter 0]

3 Apr 2023

Dear Prof. Dobnikar,

Thank you very much for submitting your manuscript "Data–driven modelling makes quantitative predictions regarding bacteria surface motility" for consideration at PLOS Computational Biology.

As with all papers reviewed by the journal, your manuscript was reviewed by members of the editorial board and by several independent reviewers. In light of the reviews (below this email), we would like to invite the resubmission of a significantly-revised version that takes into account the reviewers' comments.

We cannot make any decision about publication until we have seen the revised manuscript and your response to the reviewers' comments. Your revised manuscript is also likely to be sent to reviewers for further evaluation.

Sincerely,

Stefan Klumpp

Academic Editor

PLOS Computational Biology

Daniel Beard

Section Editor

PLOS Computational Biology

Reviewer's Responses to Questions

**Comments to the Authors:**

Reviewer #1: The manuscript by Barton et al uses computational model for twitching motility of P.aeruginosa bacteria in combination with trajectory analysis and Bayesian inference with a goal of gaining access to the parameters that are not otherwise easily accessible in experiments. We find the idea certainly useful and approach generally promising. The manuscript is mostly clearly written and introduces all the details of simulations, data analysis, and statistics rather well. Introduction and motivation of the study is also convincing. We noticed, however, that the realisation of the approach developed in the ms was hampered by several deficiencies in the study.

i) First conceptual point is that the manuscript going in the direction of the method suggestion have chosen the experimental system that does not allow to validate the proposed approach. It is true that many parameters that the authors try to restore with the help of the model still can’t be measured, but the pili dynamics for the bacteria species they are using can be visualised (and was visualised before, starting from the seminal paper of Skerker and Berg). For sure that would be much shorter time series than tracking data but that would be the crucial experimental data to validate the performance of the model simulations (which does have some limitations and apparent discrepancy with previous observations, see next point)

ii) We believe that the shortcomings on the model side might be the true reasons why the proposed parameter optimisation approach could not excel in fixing all of the parameters. Several indications that the model was not yet properly tuned include the suggestion of authors that motility is driven by short displacements of multiple pili (which seems to contradict the observations from the same Skerker and Berg paper and pili-driven motility as was imaged for Neisseria gonorrhoeae). Second point, specific to PA cells, is that in the model the cells can only move one way (walking or crawling) depending if potential is switched to attraction of repulsion while in the experiments [7] it was shown that individual cells can switch between the modes.

The rest of the work, in particular the statistical analysis and discussion of the parameters and why some of them can’t be too efficiently optimised etc. is thoroughly written.

Below we specify more concrete points of criticism, in chronological order as they appear in the text, and hoping the authors would be able to respond to those.

1) typo: in introduction, NG name should be italic

2) minor: page 5, before section A. The orientation of the cell in the model is regulated by the attraction/repulsion of the two point on the cell body. While it is clear why that is needed for the model, what is the biological explanation/basis to introduce such interaction potential?

3) minor: throughout the text the quantities are inconsistently given either with units or without, while obviously they should be always with units (if they are dimensional)

4) Major: the model is solved by energy minimisation algorithm thus disregarding the dynamics happening at short time scales. This dynamics is influenced by the viscosity of the surrounding solvent and friction with the substrate. While the authors mention that their model can reproduce the slingshot movement in their model that would correspond to immediate cell body equilibration (jump) while in reality it would be a viscosity damped process. Friction forces were suggested to be important also in the context of twitching motility (see Pönisch et al PRE 2109), while it can be absorbed in effective viscosity for purely 2D motility, in this work, 3D effects and forces in z-direction are obvious and would affect the friction force and thus the dynamics. With current approach of simple energy minimisation, those are hard to follow.

5) Major: pili retraction velocity and pili detachment rate were experimentally measured for Tfp and were shown to play central role for describing twitching motility (see Marathe et al Nat. Com. 2014 and Zaburdaev et al Biophys. J. 2014). The authors had to reduce the retraction speed (by hand so to say) to make their model produce a better fit to the data. In fact, as the attachment/detachment rates are the central parameters of the model, we feel that disregarding these effects is one of the major reasons why the model fails to give better alignment to the data (meaning more convincing matching by statistical analysis).

6) minor/Major: related to the above, strangely the pili pulling force or stalling force is actually never a parameter of the model, while biophysically this is the major quantity determining the dynamics.

7) minor: explanation of what the free pili does during interaction with a surface (bottom of page 7) - is not clear and should be rewritten

8) Major: result of the model saying that motility of cells is a result of multiple small displacements by multiple pili is not consistent with previous images/movies of twitching motility and is not critically commented in this respect.

9) Major: Section C on page 23 says that the cells in the model either crawl or walk and can’t switch their behaviour in the model, referring to [7], but also in [7] it was shown that individual cells can switch the modes of activity. So how that would be captured/explained by the model.

10) Minor: In the Appendix formula explaining body equilibration it seems that also shortening of the pilus would generate a force, while it says the opposite in the text. So maybe authors should comment on it.

Taken together, we think that model needs to be critically reassessed by the authors in the context of known pili-substrate interactions, force dependent detachment and retraction velocity, as well as the critical discussions of the observed motility with previously reported cases. Thus we consider the ms as not yet suitable for publication.

Reviewer #2: In the present manuscript, Barton et al. first set up a model for twitching motility of bacteria at surfaces, driven by retraction and extension of type-IV-pili. The model, which is based on experimental observations of this type of systems, incorporates the stochastic growth and shrinkage of pili, their reformation as well as their random distribution at one cap of the bacterium, which is modeled as a spherocylinder. Depending on the interaction potential of bacterium and surface, two modes of motion can be described: walking and crawling.

In the second part of the manuscript, a sensitivity analysis for the model parameters is performed and the most sensitive four model parameters are identified (the others are kept fixed). In the final part of the paper, the most sensitive model parameters are inferred based on tracking data of the bacterium P. aeruginosa by approximate Bayesian computation.

The authors set up the model in a very systematic way. The presentation of is in all parts clear and comprehensible. It is particularly noteworthy that parameters are systematically inferred from experimental tracking data. In this context, the authors go beyond standard approaches of fitting mean-square displacements and (velocity-) correlation functions. I therefore support the publication of this manuscript in PLOS Computational Biology.

I would like the authors, however, to take the following questions and comments into account in their revision prior to publication.

(I) trajectories, correlation functions and MSD

Though the parameter inference does not rely on standard measure, I would still encourage the authors to include those in the manuscript. First, some exemplary experimental trajectories could be shown in comparison to the simulated ones. This will help gaining some intuition on the type of random transport process.

Furthermore, a comparison of simulated and experimentally obtained mean-square displacement could now be used to compare the success/goodness of the modeling and parameter inference.

(II) trail following

There have been works, for example by Ramin Golestanian and others, discussion the question of trail following in the context of twitching motility, in particular in the context of P. aeruginosa. The model presented by the authors does not include this effect. Please clarify whether this mechanism is relevant for the understanding of the (long-time) motility of these bacteria and why it was left out in this study.

(III) simulation technique

The simulations presented by the authors rely on Monte-Carlo techniques and Boltzmann generators. These methods are designed to sample random realizations from given high dimensional distributions (oftentimes of Boltzmann type) efficiently. Depending on the specific algorithm, the resulting sequences do not necessarily reflect the temporal dynamics. In short, Monte-Carlo steps are not necessarily related to time steps. However, the comparison to experimental trajectories require that simulated data are time-ordered. Please clarify the conceptual idea of the simulation technique used here.

(IV)

There is comment (page 8) that Koch et al. made a "compatible albeit incongruous claim". This is contradicting to me and was not clear. I suggest to explain in detail what exactly is referred to here or to leave out this comment.

(V) Typos

- There is a references to Figure 1c (page 5 in my version), which should be Figure 1b.

- On the same page: "such as to balances the TFP tension..."

- There are some inconsistencies in the notation in the list of references, in particular in the capitalization of journal titles.

I saw that the simulation code is made available on github. Is the trajectory data available too?

**Have the authors made all data and (if applicable) computational code underlying the findings in their manuscript fully available?**

Reviewer #1: Yes

Reviewer #2: Yes

PLOS authors have the option to publish the peer review history of their article (what does this mean?). If published, this will include your full peer review and any attached files.

Reviewer #1: No

Reviewer #2: No
---

## [Decision Letter · Decision Letter 1]

7 Aug 2023

Dear Prof. Dobnikar,

Thank you very much for submitting your manuscript "Data–driven modelling makes quantitative predictions regarding bacteria surface motility" for consideration at PLOS Computational Biology.

As with all papers reviewed by the journal, your manuscript was reviewed by members of the editorial board and by several independent reviewers. In light of the reviews (below this email), we would like to invite the resubmission of a significantly-revised version that takes into account the reviewers' comments.

As you can see from the reports, one of the previous reviewers remains very critical about your work. We have asked a third reviewer for comments who is critical as well (with some similarity in the criticism), but also appreciates the progress made with this manuscript and makes constructive suggestions how to proceed. Please consider these suggestions in a further round of revisions.

We cannot make any decision about publication until we have seen the revised manuscript and your response to the reviewers' comments. Your revised manuscript is also likely to be sent to reviewers for further evaluation.

Sincerely,

Stefan Klumpp

Academic Editor

PLOS Computational Biology

Daniel Beard

Section Editor

PLOS Computational Biology

As you can see from the reports, one of the previous reviewers remains very critical about your work. We have asked a third reviewer for comments who is critical as well (with some similarity in the criticism), but also appreciates the progress made with this manuscript and makes constructive suggestions how to proceed. Please consider these suggestions in a further round of revisions.

Reviewer's Responses to Questions

**Comments to the Authors:**

Reviewer #1: see attachment

Reviewer #2: The authors have convincingly addressed my concern and adopted their manuscript accordingly. I therefore support publications of the manuscript, once the concerns of the second referee have been addressed.

Reviewer #3: J. Dobnikar and coworkers propose a twitching motility model for P. aeruginosa. Using published data, the pilus and cell model is parametrized and a sensitivity analysis is conducted. The authors simulate crawling motility as well as walking motility. The simulation results are compared with experimental data consisting of tracks of migrating bacteria.

The strength of this work clearly lies in the computational methodology for finding appropriate model parameters. However, the model itself is very speculative and the predictive value of the model is obviously limited by the many open questions behind the assumptions. In fairness, one can see that the authors did their best to extract as much information from the literature as possible.

Major comments:

A) Clearly, a main weakness of the work is that the model rests on a large number of assumptions that could not be tested by the authors. Examples highlighting the speculative nature of this model include

1) using the hardly established contact sensing mechanism suggested by Tala et at

2) assumed van-der-Waals-like surface attachment of the cell

3) detailed modeling the unknown process of binding between substrate and pili

4) assumed angle-dependent retraction speed

5) assumption that the switching rates measured in Koch et also hold when pili are engaging the substrate

6) assumed force-independent pilus detachment rate

7) Assumption that Brownian/active noise does not affect bacterial walking

8) Etc

Some of these assumptions are to the knowledge of the reviewer not appropriate.

Suggestion for A): Please add a comprehensive list of the assumptions, similar to Table 1, where all the assumptions are listed so that a reader can easily understand what the limits of the model are.

B) The authors emphasize the Bayesian nature of their data analysis. However, the text does not really explain well which techniques are used in practice and the provided GitHub repository does not contain the used routines. This makes it hard to understand what the authors did exactly.

Suggestion for B): since this work is really unique in its effort to use systematic parameter variation, the authors should put more effort into explaining clearly what the analysis routines are doing and provide the code.

C) There is a very confusing issue regarding stall forces of pili in this manuscript. As previous referees commented, stall forces can play a major role for the pilus dynamics. In Table I, the authors claim that the measures stall forces are 100 pN (Ribbe et al). This is wrong. Ribbe et al measure roughly 30pN. Moreover, while it is shown with sensitivity analysis that the stall forces may not matter much for migration, they may matter for the pilus statistics and retraction speed. The corresponding analysis is lacking.

Suggestion for C): please correct the given stall forces and repeat the simulations with correct value where necessary. A sensitivity analysis regarding the role of stall forces for the pili statistics should be included.

D) The authors emphasize already in the abstract their Mises-Fisher distribution for pili. “…predict previously unresolved distribution of TFP on the bacterial surface and show that

these predictions are consistent between the walking and crawling motility…”

This “prediction” is does not follow from the authors model but the functional form of the angular pili distribution an assumption. Moreover, the simulations are so complex that it is hard to see how they would confirm the validity of the chosen angular distribution. Thus the statement in the abstract is, in the reviewers opinion, misleading.

Suggestion for D): Please modify statements throughout the manuscript and replace the word “predict” with “suggest a functional form of”

E) Pole switching: It is well-known that Pseudomonads regularly switch their pole, which could affect the migration dynamics (e.g. Oliveira et al PNAS 113.23 (2016): 6532-6537.)

Suggestion for E): The authors may repeat their simulations by including reversal. How does this effect change the simulation results regarding MSDs in the authors’ work?

F) Tracking: Tracked trajectories are very variable and the authors note correctly that tracking of walking cells is challenging. One cannot avoid the sense that the authors have not invested enough care in generating the correct statistics since (i) “trajectories are selected by hand”, which amount to uncontrolled removal of outliers, (ii) no errorbars are given, (ii) only a few trajectories are used.

Suggestion for F): Ideally, the tracking should be repeated for more data without manual removal of trajectories, in particular for the walking cells. At the very minimum, the analysis should be redone without manual removal of trajectories and those results should be compared to the previous results where trajectories were removed manually. Statistics and errorbars should be provided.

G) A recent biorXiv preprint by Simsek et al. https://doi.org/10.1101/2023.05.09.538458 clarifies through experimental data and modeling most of the questions that arise in the context of this model (including pili angles, rate parameters, adhesion, etc). The authors are encouraged to discuss their model in the light of these new findings.

Minor comments:

- P. Aeruginosa -> P. aeruginosa in main text and SI

- In the SI, the authors state that they use uniform priors for their “Bayesian” analysis. However, since uniform priors are used, this is nothing but a standard parameter variation. Please clarify this in the main text and explain which aspects of Bayesian analysis are used.

-The persistence length of pili is a matter of dispute and very different quantities have been reported. While the author’s choice to use the value of 5micrometer is appealing, it would be appropriate to cite some earlier work on this topic, e.g., de Haan HW. Biophys J. 2016 Nov 15;111(10):2263-2273 or Lu, Shun, et al. Biophys. J 108.12 (2015): 2865-2875.

**Have the authors made all data and (if applicable) computational code underlying the findings in their manuscript fully available?**

Reviewer #1: Yes

Reviewer #2: **No: **The authors promised to make their tracking data available. As far as I understand, they are currently not available.

Reviewer #3: **No: **The authors have not made any of their data available. Regarding the code, only the core model code is available on GitHub, not the essential sensitivity analysis and Bayesisian analysis routines.

PLOS authors have the option to publish the peer review history of their article (what does this mean?). If published, this will include your full peer review and any attached files.

Reviewer #1: No

Reviewer #2: No

Reviewer #3: No
---

## [Decision Letter · Decision Letter 2]

26 Jan 2024

Dear Prof. Dobnikar,

Thank you very much for submitting your manuscript "Data–driven modelling makes quantitative predictions regarding bacteria surface motility" for consideration at PLOS Computational Biology.

As with all papers reviewed by the journal, your manuscript was reviewed by members of the editorial board and by several independent reviewers. In light of the reviews (below this email), we would like to invite the resubmission of a significantly-revised version that takes into account the reviewers' comments.

As you can see from the reports, both reviewers remain unconvinced by some of your answers and still raise important critical points that you should seriously consider. I reread your manuscript and I think that its strength is in the method with which parameters are extracted from data, while a weakness is that the model depends on assumptions that may or may not be valid, at least some seem to be controversial. Therefore, the critical discussion of the model assumptions that both reviewer asked for, already in the previous round, is clearly warranted in my opinion. I send the manuscript back to you for a last round of revision, after which I hope to make a final decision.

We cannot make any decision about publication until we have seen the revised manuscript and your response to the reviewers' comments. Your revised manuscript is also likely to be sent to reviewers for further evaluation.

Sincerely,

Stefan Klumpp

Academic Editor

PLOS Computational Biology

Daniel Beard

Section Editor

PLOS Computational Biology

As you can see from the reports, both reviewers remain unconvinced by some of your answers and still raise important critical points that you should seriously consider. I reread your manuscript and I think that its strength is in the method with which parameters are extracted from data, while a weakness is that the model depends on assumptions that may or may not be valid, at least some seem to be controversial. Therefore, the critical discussion of the model assumptions that both reviewer asked for, already in the previous round, is clearly warranted in my opinion. I send the manuscript back to you for a last round of revision, after which I hope to make a final decision.

Reviewer's Responses to Questions

**Comments to the Authors:**

Reviewer #1: I reread the manuscript from the scratch. In order not to go into another round of back and forth with the authors, I think I can state that the current version of the manuscript provides clearly the details of how simulations are done and how the model is used to infer the twitching motility characteristics. I still believe that the model is not critically discussed with respect to the assumptions made. Instead we mostly presented with how good it matches the experimental data. I can't understand why the authors are so reluctant to do that critical assessment, given that it was never a barrier for this work to be published. For example, a suggestion of the Reviewer 3 with the table of assumptions would work, or a subsection doing the same. Being critical and discussing limitations is what stimulates further research and leads towards novel discoveries. All that being said, I also think that a somewhat informed reader would be able to recognize the limitations of the model by carefully reading this manuscript, and would be able to judge the reliability of the results or to adjust/modify the method for their own use/data. Therefore I don't want to use our arguments from previous rounds of review to stop this work from being published and I leave it up to editors if a concise critical discussion of the model limitations need to be added to the manuscript.

Reviewer #3: The authors have made some effort to respond to the comments from the reviewer. However, some aspects remain unsatisfactory and really should be fixed so that the otherwise interesting work can be better appreciated by the community.

1. The authors now use the very low value of retraction speed for anchored pili measured in Zhang et al. Readers may buy that this is appropriate for the experiments used as reference for this work. However, the experiments are almost not explained in the main text. Do the authors use the data from Ni et al.? Regardless where the data comes form, the experiments need to be explained in detail since only that will allow the reader to understand the assumptions. At which temperature were the experiments conducted that the authors analyzed? What was the setup? The microscope? Which medium was used? Were the experiments conducted in Log-phase? etc etc etc

2. The authors seem to have a lot of confidence in the relevance of a "tip-sensing mechanism" suggested by Tala et al. Although they admit that it seems hard to prove tip sensing, the authors now add a cryptic sentence claiming that tip sensing is necessary in their simulations. Without substantial evidence, including data and simulation analysis, such a statement makes no sense in the Reviewer's opition. Please either provide that evidence or remove the statement.

Please note that Tala et al. could only analyze extremely long pili due to imaging artifacts caused by the cell body. The delay times measured for the "rare-event long pili" are hardly representative of what's really going on. Tip sensing has neither been reported in extensive previous work with optical traps, nor in later work with labeled pili. Thus, it must be seen as highly speculative.

3. The author's didn't bother to respond to the question whether Brownian noise affects the motion of cells. While it is clear that tightly adherent cells are not affect by this noise, walking bacteria may transiently detach from the surface. Such jumps may strongly affect the nature of the diffusive random walk. Please analyze the data regarding this point and provide a conclusive statement on whether this is relevant for your experiments.

4. Since Reviewer 1 seems to share my concerns regarding the intransparency of the model, I would like to repeat my suggestion regarding a comprehensive table summarizing the model assumptions to the main text. I really feel that this table would improve the transparency of the work.

**Have the authors made all data and (if applicable) computational code underlying the findings in their manuscript fully available?**

Reviewer #1: Yes

Reviewer #3: **No: **There is some simulation code but I cannot ascertain that code for the bayesian is included.

PLOS authors have the option to publish the peer review history of their article (what does this mean?). If published, this will include your full peer review and any attached files.

Reviewer #1: No

Reviewer #3: No
---

## [Editor Report · Decision Letter 3]

9 Apr 2024

Dear Prof. Dobnikar,

We are pleased to inform you that your manuscript 'Data–driven modelling makes quantitative predictions regarding bacteria surface motility' has been provisionally accepted for publication in PLOS Computational Biology.

Best regards,

Stefan Klumpp

Academic Editor

PLOS Computational Biology

Daniel Beard

Section Editor

PLOS Computational Biology

To save time, as this manuscript has been under review for quite some time now, I have not sent it back to reviewers, but re-read the revision myself. As stated already for an earlier version, I see a main strength of this work in the analysis method, while the details of the modeling done here have led to extended discussions with some of the reviewers. In my opinion, the authors have clarified all remaining issues. in particular, they now spell out assumptions explicitly and list them in a table as suggested by the reviewers. This is as transparent as possible and further discussions can and should be based on the published paper. It is also good to see that the tip sensing mechanism (which seems to be somewhat controversial) is not crucial for the results.

---

## [Editor Report · Acceptance letter]

9 May 2024

PCOMPBIOL-D-22-01843R3 

Data–driven modelling makes quantitative predictions regarding bacteria surface motility

Dear Dr Dobnikar,

I am pleased to inform you that your manuscript has been formally accepted for publication in PLOS Computational Biology. Your manuscript is now with our production department and you will be notified of the publication date in due course.

With kind regards,

Anita Estes
